# Spectral Graph Coarsening Using Inner Product Preservation and the Grassmann Manifold

**Ido Cohen**
Electrical and Computer Engineering
Technion–Israel Institute of Technology
sidoc@campus.technion.ac.il

**Ronen Talmon**
Electrical and Computer Engineering
Technion–Israel Institute of Technology
ronen@ee.technion.ac.il

## Abstract

We propose a novel functorial graph coarsening method that preserves inner products between node features, a property often overlooked by existing approaches focusing primarily on structural fidelity. By treating node features as functions on the graph and preserving their inner products, our method retains both structural and feature relationships, facilitating substantial benefits for downstream tasks. To formalize this, we introduce the Inner Product Error (IPE), which quantifies how the inner products between node features are preserved. Leveraging the underlying geometry of the problem on the Grassmann manifold, we formulate an optimization objective that minimizes the IPE, also for unseen smooth functions. We show that minimizing the IPE improves standard coarsening metrics, and illustrate our method's properties through visual examples that highlight its clustering ability. Empirical results on benchmarks for graph coarsening and node classification show that our approach outperforms existing state-of-the-art methods.

## 1 Introduction

Graph-structured data has become ubiquitous in a wide range of domains, including social networks [1], biological systems [2], and recommendation systems [3], due to its ability to model complex relationships and interactions. With the exponential increase in data availability, the size of graphs in many applications has also grown significantly. This surge in graph size presents major challenges, as traditional and even advanced graph processing techniques often become computationally infeasible or excessively time-consuming when applied to large-scale graphs. To address these issues, graph reduction techniques are developed with the aim of simplifying large graphs while retaining key structural features, thereby enhancing computational efficiency. There are three main strategies for graph reduction [4]: graph sparsification, graph condensation, and graph coarsening.

Graph sparsification [5, 6] reduces graph size by selectively removing edges and nodes while maintaining overall structural properties. However, there is a limit to how much a graph can be sparsified without compromising its integrity. Graph condensation methods [7, 8] aim to reduce graph size by generating a smaller, synthetic graph that replicates the performance of the original graph on specific tasks, such as training a Graph Neural Network (GNN). Although condensation significantly lowers computational costs, it may not retain a clear structural interpretation, making it difficult to understand how or why certain nodes or edges are represented in the reduced version.

In contrast, graph coarsening [9] is a traditional approach that reduces graph size by grouping similar nodes into super-nodes, aiming to approximate the original structure. These methods typically seek to preserve key structural properties such as spectral characteristics [10], connectivity [11], and community structure [12, 13]. Most existing coarsening methods focus primarily on structure and often overlook node features, which play a critical role in many graph learning tasks. These methods typically operate solely on the graph topology, neglecting the rich information encoded in the node

39th Conference on Neural Information Processing Systems (NeurIPS 2025).

features. Recently, the Featured Graph Coarsening (FGC) method was proposed to address this limitation by incorporating node features into the coarsening process [14]. FGC emphasizes the reconstruction of the node features after coarsening and promotes smoothness in the coarsened graph as part of its optimization objective. However, since FGC focuses on preserving the individual characteristics of the node features, it does not fully exploit their mutual relationships, which may encode valuable information.

In this work, we propose a new approach to graph coarsening from a functorial perspective. We treat node features as functions, or signals, defined on the graph, focusing on preserving the relationships between these functions by maintaining their inner products during coarsening. Our method introduces a new coarsening metric, the Inner Product Error (IPE), which measures how the inner products between graph signals are preserved. We postulate that minimizing IPE ensures that the coarsened graph retains structural consistency and node feature relationships crucial for graph learning tasks. We exploit the geometry of the problem by recognizing that both the coarsening operator and the matrix spanning the node features (under a smoothness assumption) can be viewed as points on the Grassmann manifold. Leveraging the properties of the Grassmann manifold, we extend IPE minimization beyond observed node features, enabling our method to generalize to unseen features that satisfy the smoothness assumption. This approach is formulated as an optimization problem, and we compute the coarsening operator using gradient descent. Additionally, we theoretically show that minimizing our proposed approach leads to improvements in common graph coarsening metrics.

To demonstrate the effectiveness of our method, we present visual and empirical results showing that, while our method focuses on the functional relationships between node features, it also captures the global structure of the graph. We validate its performance through extensive experiments on multiple graph coarsening and node classification benchmarks, where our method consistently outperforms state-of-the-art coarsening methods, demonstrating its practical utility.

## 2    Background

**Grassman manifold**    The set of $n \times k$ matrices whose columns are orthonormal vectors forms a Riemannian manifold called the Stiefel manifold [15] defined by,

$$\text{St}(n, k) := \{\boldsymbol{U} \in \mathbb{R}^{n \times k} | \boldsymbol{U}^T \boldsymbol{U} = \boldsymbol{I}_{k \times k}\}. \tag{1}$$

where $\boldsymbol{I}_{k \times k}$ is a rank-$k$ identity matrix. The Grassmann manifold $\text{Gr}(n, k)$ is a quotient manifold representing the set of $k$-dimensional subspaces of the Euclidean space $\mathbb{R}^n$. Two points on the Stiefel manifold that span the same subspace represent the same point on the Grassmann manifold [16, 17]. In general, a point on $\text{Gr}(n, k)$ is represented by an equivalence class

$$[\boldsymbol{U}] = \{\boldsymbol{U}\boldsymbol{O} : \boldsymbol{O} \in \mathcal{SO}(k)\}, \tag{2}$$

where $\boldsymbol{U} \in \text{St}(n, k)$, and $\mathcal{SO}(k)$ are all $k \times k$ rotation matrices (also known as the special orthogonal group), such that any $\boldsymbol{O} \in \mathcal{SO}(k)$ satisfies $\boldsymbol{O}\boldsymbol{O}^T = \boldsymbol{O}^T\boldsymbol{O} = \boldsymbol{I}_{k \times k}$. The principal angles between two subspaces $\boldsymbol{U}_1$ and $\boldsymbol{U}_2$ are the angles that measure the smallest angular separation between basis vectors in one subspace and basis vectors in the other subspace. We denote them by $\boldsymbol{\theta} = [\theta^{(1)}, \theta^{(2)}, \dots, \theta^{(k)}]$. Given two subspaces $\boldsymbol{U}_1$ and $\boldsymbol{U}_2$ on the Grassmann manifold $\text{Gr}(n, k)$, the cosine of the principal angles between them can be computed using the SVD decomposition of $\boldsymbol{U}_1^T \boldsymbol{U}_2 = \boldsymbol{A}\boldsymbol{\Theta}\boldsymbol{B}^T$, where the singular values on the diagonal of $\boldsymbol{\Theta}$ are $[\cos(\theta^{(1)}), \cos(\theta^{(2)}), \dots, \cos(\theta^{(k)})]$.

The geodesic similarity on the Grassmann manifold is defined using the principal angles between subspaces [17]. The authors in [18] showed that this geodesic similarity can be computed by,

$$Gr(\boldsymbol{U}_1, \boldsymbol{U}_2) = \sum_{i=1}^{k} \cos^2(\theta^{(i)}) = tr(\boldsymbol{U}_1\boldsymbol{U}_1^T\boldsymbol{U}_2\boldsymbol{U}_2^T). \tag{3}$$

**Graph coarsening**    A graph with node features is denoted by the quadruplet $\mathcal{G} = (\mathcal{V}, \mathcal{E}, \boldsymbol{W}, \boldsymbol{X})$, where $\mathcal{V}$ is a set of $n$ vertices, $\mathcal{E}$ is a set of edges, $\boldsymbol{W} \in \mathbb{R}^{n \times n}$ is a weighted adjacency matrix, and $\boldsymbol{X} \in \mathbb{R}^{n \times p}$ is a node features matrix such that each row specifies the values of the $p$ features for each node. Each node feature, represented as a column of $\boldsymbol{X}$, can also be considered as a graph signal $\boldsymbol{x} \in \mathbb{R}^n$, assigning a real value to each vertex, namely, $\boldsymbol{x} : \mathcal{V} \to \mathbb{R}$. The graph Laplacian matrix $\boldsymbol{L}$ is

defined by $L = D - W$, where $D = \text{diag}(W\mathbf{1})$ is the diagonal degree matrix. We define the inner product between two graph signals $x, y \in \mathbb{R}^n$ with respect to the graph $\mathcal{G}$ by:

$$\langle x, y \rangle_L = x^T L y = \sum_{(i,j) \in E} w_{ij}(x(i) - x(j))(y(i) - y(j)), \tag{4}$$

where $w_{ij}$ are edge weights, and $x(i), y(i)$ are the values of the features at node $i$. Since $L$ is a positive semi-definite matrix, $x^\top L x$ induces a semi-norm and defines an inner product on the subspace of $\mathbb{R}^n$ orthogonal to the constant vector $\mathbf{1}$, as discussed in Von Luxburg [19]. We denote the graph Laplacian eigen decomposition by $L = U \Lambda U^T$, where the columns of $U$ are the eigenvectors of $L$, and $\Lambda$ is a diagonal matrix consisting of its corresponding eigenvalues.

Given a graph $\mathcal{G} = (\mathcal{V}, \mathcal{E}, W, X)$ with $n$ nodes, the goal of graph coarsening is to construct a coarsened graph $\mathcal{G}_c = (\mathcal{V}_c, \mathcal{E}_c, W_c, X_c)$ with $k \ll n$ nodes, while preserving the main structural properties of $\mathcal{G}$, thereby simplifying subsequent analysis and computations. The coarsening procedure is defined through a linear mapping $\pi : \mathcal{V} \to \mathcal{V}_c$ that maps nodes in $\mathcal{G}$ to nodes in $\mathcal{G}_c$, termed 'super-nodes'. This linear mapping is defined by the coarsening matrix $P \in \mathbb{R}_+^{k \times n}$, such that $X_c = PX$. Each non-zero entry of $P$ indicates a mapping from a node in $\mathcal{G}$ to a super-node in $\mathcal{G}_c$, i.e., if and only if the $j$-th node in $\mathcal{G}$ is mapped to the $i$-th super-node of $\mathcal{G}_c$, then $P_{i,j} > 0$. Let $L \in \mathbb{R}^{n \times n}$ and $L_c \in \mathbb{R}^{k \times k}$ be the respective Laplacian matrices of $\mathcal{G}$ and $\mathcal{G}_c$, and let $L_l \in \mathbb{R}^{n \times n}$ and $X_l \in \mathbb{R}^{n \times p}$ be the lifted Laplacian and feature matrices, i.e., the reconstructed full graph matrices after the coarsening procedure. The relationships between the coarse graph Laplacian and features and the original graph Laplacian and features are [20]:

$$L_c = C^\top L C, \qquad\qquad X_c = PX \tag{5}$$
$$L_l = P^\top L_c P, \qquad\qquad X_l = C X_c \tag{6}$$

where $C \in \mathbb{R}_+^{n \times k}$ is the pseudo-inverse of $P$, i.e., $C = P^\dagger$. The non-zero entries of $C$ also imply a node mapping from $\mathcal{G}$ to $\mathcal{G}_c$, such that $C_{i,j} > 0$ if the $i$-th node of $\mathcal{G}$ is mapped to the $j$-th super-node of $\mathcal{G}_c$. We note that the matrix $C$ belongs to the following set:

$$\mathcal{C} = \{C \geq 0 | \langle C_{:,i}, C_{:,j} \rangle = 0 \quad \forall i \neq j, \tag{7}$$
$$\langle C_{:,i}, C_{:,i} \rangle = d_i, \|C_{:,i}\|_0 \geq 1, \|C_{i,:}\|_0 = 1\}$$

where $C_{:,i}$ is the $i$-th orthogonal column of $C$, $C_{i,:}$ is the $i$-th row of $C$, $\langle \cdot, \cdot \rangle$ is the standard inner product, and $d_i$ is some positive number. Since the columns of a valid coarsening matrix $C$ are orthogonal, it can also be viewed as a point on the Grassmann manifold $\text{Gr}(n, k)$.

Numerous evaluation metrics exist for graph coarsening, each assessing how well specific graph properties are preserved during reduction. Next, we review the main ones.

**Definition 2.1 (Relative Eigen Error (REE) [10])** *The REE is defined as $REE = \frac{1}{k} \sum_{i=1}^{k} \frac{\lambda_{c,i} - \lambda_i}{\lambda_i}$, where $\lambda_i$ and $\lambda_{c,i}$ are the $k$ dominant eigenvalues of the original graph Laplacian matrix $L$ and the coarsened graph Laplacian matrix $L_c$, respectively.*

**Definition 2.2 (Reconstruction Error (RE) [21])** *The RE between the original graph Laplacian $L$ and the lifted graph Laplacian $L_l$ is defined by $RE = \|L - L_l\|_F^2$.*

The REE and RE are coarsening metrics independent of the graphs' node features. The REE measures the spectral similarity between graphs and how global properties such as important edges are preserved, while the RE quantifies how local information is preserved during coarsening.

**Definition 2.3 (Hyperbolic Error (HE) [22])** *The HE between the original Laplacian matrix $L$ and lifted Laplacian matrix $L_l$ is defined as $HE = \text{arccosh}\left(1 + \frac{\|(L - L_l)X\|_F^2 \|X\|_F^2}{2\text{tr}(X^T L X)\text{tr}(X^T L_l X)}\right)$, where $X$ is the node features matrix of the original graph.*

**Definition 2.4 (Dirichlet Energy Error (DEE))** *The Dirichlet Energy (DE) of a graph is defined by $DE = \text{tr}(X^\top L X)$, where $L$ denotes the graph Laplacian and $X$ denotes the node feature matrix of the graph [23]. We define the DEE between the original graph $\mathcal{G}$ and its coarsened version $\mathcal{G}_c$ as $DEE = \left|\log\left(\frac{DE_\mathcal{G}}{DE_{\mathcal{G}_c}}\right)\right|$, where $DE_\mathcal{G}$ and $DE_{\mathcal{G}_c}$ are the DE of the original and coarsened graphs.*

The HE and DEE are coarsening measures that consider the node features. The HE measures distortion in the geometric structure of the data in hyperbolic space. This is useful when the graph

has a hierarchical structure (e.g., trees), as hyperbolic spaces are suited for representing such data. The DE measures the smoothness of the node features on a graph; lower DE values suggest that the node features are closely aligned with the graph structure. Consequently, we define the DEE which quantifies the extent to which the intrinsic graph structure in the node features is preserved during the coarsening process. We note that the authors in Kumar et al. [14] suggest minimizing the DE of the coarsened graph as part of their graph coarsening optimization objective.

## 3 Proposed method

Our method adopts a functorial perspective for graph coarsening, focusing on maintaining the relationships between different functions defined on the graph. Specifically, it aims to preserve the inner products between functions defined on the graph. To achieve this, we first introduce the following new graph coarsening metric that quantifies how the inner products between given graph signals (i.e., node features) are preserved during the coarsening process.

**Definition 3.1 (Inner Product Error (IPE))** *Let $\boldsymbol{L}$ and $\boldsymbol{X}$ be the original graph Laplacian and node features matrix, and let $\boldsymbol{L}_c$ and $\boldsymbol{X}_c$ be their respective coarsened graph Laplacian and features matrix. The Inner Product Error (IPE) is defined by IPE $= \|\boldsymbol{X}^\top \boldsymbol{L} \boldsymbol{X} - \boldsymbol{X}_c^\top \boldsymbol{L}_c \boldsymbol{X}_c\|_F^2$.*

The motivation for this approach is based on the following:

**Proposition 3.2** *Let $\boldsymbol{L}$ and $\boldsymbol{L}_c$ be the graph Laplacians of a graph $\mathcal{G}$ and its coarsened graph $\mathcal{G}_c$, respectively. If for all pairs of graph signals $\boldsymbol{x}, \boldsymbol{y} \in \mathbb{R}^n$, the inner product between the two signals is preserved under the coarsening process, i.e.:*

$$\boldsymbol{x}^\top \boldsymbol{L} \boldsymbol{y} = \boldsymbol{x}_c^\top \boldsymbol{L}_c \boldsymbol{y}_c,$$

*then the graph Laplacian of the original graph, $\boldsymbol{L}$, can be fully reconstructed from $\boldsymbol{L}_c$ via:*

$$\boldsymbol{L} = \boldsymbol{L}_l = \boldsymbol{P}^\top \boldsymbol{L}_c \boldsymbol{P}$$

See App. A.1 for proof. Prop. 3.2 shows that preserving inner products of graph signals maintains key structural properties, enabling graph reconstruction. We note that a necessary condition for the assumption in Proposition 3.2 to hold is that $\text{rank}(\boldsymbol{L}) < k$, which implies that the graph has at least $n - k$ connected components, a condition that is rarely met in practice. Yet, we show in Sec. 4 that our approach achieves the lowest reconstruction error (RE) among baselines, even when this criterion is violated, demonstrating its broad applicability. In Sec. 3.1, we provide analytical evidence that minimizing IPE leads to the minimization of other coarsening metrics as well.

**Preserving inner products using the Grassmann manifold.** We propose an optimization approach for graph coarsening to preserve inner products between graph signals. The objective function and constraints are given by:

$$\min_{\boldsymbol{C}} f(\boldsymbol{C}) = \|\boldsymbol{X}^\top \boldsymbol{L} \boldsymbol{X} - \boldsymbol{X}_c^\top \boldsymbol{L}_c \boldsymbol{X}_c\|_F^2 - \beta Gr(\boldsymbol{C}, \boldsymbol{U}^{(k)}) + \lambda g(\boldsymbol{C}) + \alpha h(\boldsymbol{L}_c) \qquad (8)$$

$$\textbf{s.t.} \quad \boldsymbol{L}_c = \boldsymbol{C}^T \boldsymbol{L} \boldsymbol{C}, \boldsymbol{X}_c = \boldsymbol{C}^\dagger \boldsymbol{X}, \boldsymbol{C} \in \mathcal{C}$$

where $\boldsymbol{C} \in \mathbb{R}^{n \times k}$ is the coarsening operator; $\boldsymbol{L} \in \mathbb{R}^{n \times n}$ and $\boldsymbol{X} \in \mathbb{R}^{n \times p}$ are the given graph Laplacian and feature matrix of the original graph; $\boldsymbol{U}^{(k)}$ is a matrix containing the $k$ leading eigenvectors of $\boldsymbol{L}$; $\boldsymbol{L}_c \in \mathbb{R}^{k \times k}$ and $\boldsymbol{X}_c \in \mathbb{R}^{k \times p}$ are the Laplacian and feature matrix of the coarsened graph, and $\mathcal{C}$ is the set defined in (7). The function $Gr(\cdot, \cdot)$ is the Grassmann similarity score defined in (3), and functions $h(\cdot)$ and $g(\cdot)$ are regularization functions for $\boldsymbol{L}_c$ and $\boldsymbol{C}$, while $\lambda, \alpha > 0$ are positive regularization parameters.

The objective in (8) minimize the IPE using two complementary terms. The first involves directly minimizing the IPE on the given node features. Although this improves performance on the available data, it does not generalize well to new signals, as its effectiveness depends heavily on the specific information encoded in the feature matrix $\boldsymbol{X}$. The second term aims to maximize the Grassmann similarity (3) between the coarsening matrix $\boldsymbol{C}$ and the leading eigenvectors $\boldsymbol{U}^{(k)}$ of $\boldsymbol{L}$. In Sec. 3.1, we show analytically that this alignment promotes IPE minimization for general unseen smooth signals, e.g., satisfying a smoothness assumption (Prop. 3.4), and preserves important structural properties (Prop. 3.5). The parameter $\beta$ balances the two terms, adjusting the emphasis between performance on the observed data and generalization to new signals. Our empirical results show that both terms contribute to the coarsening process.

**Proposed algorithms.** One limitation of the objective in (8) is that the derivative of the first term does not have a closed-form expression with respect to $\boldsymbol{C}$. As a remedy, we adopt the multi-block optimization framework suggested by Kumar et al. [14], recasting our objective function as:

$$\min_{\boldsymbol{C}} f(\boldsymbol{X}_c, \boldsymbol{C}) = \|\boldsymbol{X}^\top \boldsymbol{L} \boldsymbol{X} - \boldsymbol{X}_c^\top \boldsymbol{C}^\top \boldsymbol{L} \boldsymbol{C} \boldsymbol{X}_c\|_F^2 - \beta tr(\boldsymbol{U}^{(k)}(\boldsymbol{U}^{(k)})^\top \boldsymbol{C} \boldsymbol{C}^\top) \tag{9}$$

$$+ \lambda \|\boldsymbol{C}^T\|_{1,2}^2 - \alpha \log \det(\boldsymbol{C}^\top \boldsymbol{L} \boldsymbol{C} + \boldsymbol{J})$$

$$\text{s.t.} \quad \boldsymbol{X}_c = \boldsymbol{C}^\dagger \boldsymbol{X}, \quad \boldsymbol{C} \in \mathcal{C}$$

where the terms involving Grassmann similarity and $\boldsymbol{L}_c$ are written explicitly. We adopt the same regularization functions as in Kumar et al. [14], which promote balanced super-node assignment and connectivity in the coarsened graph. The function $g(\boldsymbol{C}) = \|\boldsymbol{C}^\top\|_{1,2}^2 = \sum_{i=1}^n \left(\sum_{j=1}^k |C_{i,j}|\right)^2$ is an $l_{1,2}$-based group penalty that, as shown in [24, 14], encourages valid coarsening operators. The second regularization term is $h(\boldsymbol{L}_c) = \log \det(\boldsymbol{L}_c + \boldsymbol{J})$, where $\boldsymbol{J} = \frac{1}{k} \boldsymbol{1}_{k \times k}$. This term ensures that $\boldsymbol{L}_c + \boldsymbol{J}$ is full rank, implying $\text{rank}(\boldsymbol{L}_c) = k - 1$, which guarantees that the coarsened graph $\mathcal{G}_c$ is connected [25, 26].

In this recast, the derivative of the modified objective function has a closed-form expression with respect to $\boldsymbol{C}$, and the gradient is presented in App. A.4. We optimize this objective by applying projected gradient descent [27] to estimate the matrix $\boldsymbol{C}$. Specifically, applying ordinary gradient descent steps could deviate from the feasible set $\mathcal{C}$ on the Grassmann manifold. Therefore, we use projected gradient descent to periodically project $\boldsymbol{C}$ back onto the Grassmann manifold after a fixed number of gradient descent steps using the operator Hardmax($\boldsymbol{C}$). Hardmax($\boldsymbol{C}$) applies a hard maximum at each row of $\boldsymbol{C}$, ensuring column-wise orthogonality, which coincides with the structural constraints of the Grassmann manifold.

A full description of this method is in Algorithm 1, termed INGC. In Algorithm 2, we present another version, where we omit the first term in (9). In this case, we denote the algorithm's objective function by $\tilde{f}(\boldsymbol{C})$. This version focuses on minimizing the IPE via Grassmann similarity, removes dependence on the feature matrix $\boldsymbol{X}$, and enables estimating $\boldsymbol{C}$ using standard gradient descent. We refer to this algorithm as We note that, for SINGC, we empirically observed that projecting only once at the end of the optimization yields performance similar to using intermediate projections, as done for INGC. Therefore, for simplicity and efficiency, we chose to apply standard gradient descent throughout the iterations and perform the projection using Hardmax($\boldsymbol{C}$) only at the end.

| **Algorithm 1** INGC Algorithm | **Algorithm 2** SINGC Algorithm |
|---|---|
| **Input:** $\boldsymbol{L} \in \mathbb{R}^{n \times n}, \boldsymbol{X} \in \mathbb{R}^{n \times p}$ 
 **Parameters:** $\beta, \lambda, \alpha, \eta, t_{iter}, c_{iter}$ 
 **Output:** $\boldsymbol{L}_c \in \mathbb{R}^{k \times k}, \boldsymbol{X}_c \in \mathbb{R}^{k \times p}$ | **Input:** $\boldsymbol{L} \in \mathbb{R}^{n \times n}$ 
 **Parameters:** $\lambda, \alpha, \eta, t_{iter}$ 
 **Output:** $\boldsymbol{L}_c \in \mathbb{R}^{k \times k}$ |
| 1: Compute $\boldsymbol{U}^{(k)}$. | 1: Compute $\boldsymbol{U}^{(k)}$. |
| 2: Initialize $\boldsymbol{C}_0, t = 0$. | 2: Initialize $\boldsymbol{C}_0, t = 0$. |
| 3: **while** $\|\boldsymbol{C}_{t+1} - \boldsymbol{C}_t\|_F < \epsilon_C$ or $t < t_{iter}$ **do** | 3: **while** $\|\boldsymbol{C}_{t+1} - \boldsymbol{C}_t\|_F < \epsilon_C$ or $t < t_{iter}$ **do** |
| 4: $\quad \boldsymbol{C}_{t(0)} = \boldsymbol{C}_t$. | |
| 5: $\quad$ Compute $\nabla_{\boldsymbol{C}} f(\boldsymbol{X}_c, \boldsymbol{C})$ (see (16)) | 4: $\quad$ Compute $\nabla_{\boldsymbol{C}} \tilde{f}(\boldsymbol{C})$ (see (17)) |
| 6: $\quad$ **for** $i$ in range($c_{iter}$) **do** | |
| 7: $\quad\quad$ Update $\boldsymbol{C}_{t+1}$ using the gradient descent step: 
 $\quad \boldsymbol{C}_{t(i+1)} \leftarrow \boldsymbol{C}_{t(i)} - \eta \nabla_{\boldsymbol{C}} f(\boldsymbol{X}_{c(t)}, \boldsymbol{C}_{t(i)})$ | 5: $\quad$ Update $\boldsymbol{C}_{t+1}$ using the gradient descent step: 
 $\quad \boldsymbol{C}_{t+1} \leftarrow \boldsymbol{C}_t - \eta \nabla_{\boldsymbol{C}} \tilde{f}(\boldsymbol{C}_t)$ |
| 8: $\quad$ **end for** | |
| 9: $\quad \boldsymbol{C}_{t+1} = \text{Hardmax}(\boldsymbol{C}_{t(c_{iter})})$ | 6: $\quad t = t + 1$ |
| 10: $\quad \boldsymbol{X}_{c(t+1)} = \boldsymbol{C}_{t+1}^\dagger X, \quad t = t + 1$ | |
| 11: **end while** | 7: **end while** |
| 12: Hardmax(C) | 8: Hardmax(C) |
| 13: **return** $\boldsymbol{L}_c = \boldsymbol{C}_t^\top \boldsymbol{L} \boldsymbol{C}, \boldsymbol{X}_c = \boldsymbol{C}_t^\dagger \boldsymbol{X}$ | 9: **return** $\boldsymbol{L}_c = \boldsymbol{C}_t^\top \boldsymbol{L} \boldsymbol{C}$ |

## 3.1 Theoretical analysis

Next, we present two key analytical results. First, we show that the second term in (8) minimizes the IPE for general smooth graph signals. Second, we connect this minimization and common graph coarsening metrics, such as DEE and REE. We begin by defining a smooth graph signal. Dong et al. [28] model a smooth graph signal generation mechanism as:

$$x = Uh + \epsilon_\eta \eta, \tag{10}$$

where $L = U\Lambda U^\top$ is the Laplacian of the respective graph signal, $h \sim \mathcal{N}(0, \Lambda^\dagger) \in \mathbb{R}^n$, $\eta \sim \mathcal{N}(0, I_{n \times n}) \in \mathbb{R}^n$, and $\epsilon_\eta > 0$ is the noise standard deviation. This model suggests that a smooth graph signal is a combination of the first eigenvectors of $L$ and scaled noise.

**Assumption 3.3** ($k$-smooth graph signal [29]) *A graph signal $x \in \mathbb{R}^n$ is termed "$k$-smooth" on the graph $\mathcal{G}$ if it can be fully expressed by the first $k$ eigenvectors of its corresponding Laplacian $L$, i.e., $x = \sum_{i=1}^{k} c_i u^{(i)} = c(U^{(k)})^\top$ where the columns of $U^{(k)} = [u^{(1)}, \ldots, u^{(k)}] \in \mathbb{R}^{n \times k}$ are the $k$-leading eigenvectors of $L$.*

Table 4 in App. E shows the extent to which this assumption holds in real datasets. Based on Assumption 3.3, the following result provides motivation to incorporate the Grassmann similarity score $Gr(C, U^{(k)})$ defined in (3) into our proposed objective.

**Proposition 3.4** *Let $X$ be a feature matrix of a graph $\mathcal{G}$ with a Laplacian matrix $L$, where each column of $X$ is $k$-smooth on the graph $\mathcal{G}$. Then, any mapping $L_c = CLC^\top$, $X_c = C^\top X$, such that $C = U^{(k)}O$ satisfies:*

$$\|X^\top LX - X_c^\top L_c X_c\|_F^2 = 0$$

*where the columns of $U^{(k)} \in \mathbb{R}^{n \times k}$ are the $k$-leading eigenvectors of $L$, and $O \in \mathcal{SO}(k)$.*

See App. A.2 for proof. Proposition 3.4 implies that any coarsening operator $C$ whose columns span the same subspace as $U^{(k)}$ minimizes the IPE in (3.1) for any $k$-smooth signals on the original graph. Thus, the second term in our objective (8) maximizes the Grassmann similarity (3) between $C$ and $U^{(k)}$, aiming to find a valid coarsening operator (i.e., $C \in \mathcal{C}$) that satisfies $C = U^{(k)}O$.

Next, we present a theorem that provides bounds on the DEE (Def. 2.4) and REE (Def. 2.1) as functions of $\epsilon$, which quantifies the deviation of the second term in our objective from its optimal value (if $C$ and $U^{(k)}$ span the same subspace, then $\text{tr}(U^{(k)}(U^{(k)})^\top CC^\top) = k$).

**Proposition 3.5** *Let $L$ be the Laplacian of a connected graph $\mathcal{G}$, and let $U^{(k)}$ be the matrix containing its $k$-leading eigenvectors. Suppose $L_c = C^\top LC$ is the Laplacian of a coarsened graph derived using a coarsening operator $C$ such that $\text{tr}(U^{(k)}(U^{(k)})^\top CC^\top) = k - \epsilon$, and the constant vector in $\mathbb{R}^n$ is spanned by the columns of $C$. Then, the eigenvalues of the original graphs $\{\lambda^i\}_{i=1}^k$ and the coarsened graph $\{\lambda_c^i\}_{i=1}^k$ satisfy,*

$$\frac{1}{\mu_1}\lambda^{(i)} \leq \lambda_c^{(i)} \leq \frac{1}{\mu_k} \frac{(1 + \epsilon\kappa)^2}{1 - (\epsilon\kappa)^2(\lambda^{(i)}/\lambda^{(2)})} \lambda^{(i)}, \quad 2 \leq i \leq k$$

*and the Dirichlet energies of a $k$-smooth graph signal $x$ on the graph, the inequalities:*

$$(1 - \epsilon\kappa)^2 \|x\|_L \leq \|x_c\|_{L_c} \leq (1 + \epsilon\kappa)^2 \|x\|_L,$$

*whenever $\epsilon\kappa < \frac{\lambda^{(2)}}{\lambda^{(i)}}$. Here $\kappa = \frac{\lambda_{max}(L)}{\lambda^{(2)}}$, $\lambda_{max}(L)$ is the maximum eigenvalue of $L$, $P = C^\dagger$, $\mu_1, \mu_k$ and the first and $k$ eigenvalues of the matrix $PP^\top$, $\|x\|_L = x^T Lx$, $\|x_c\|_{L_c} = x_c^T L_c x_c$ .*

See App.A.3 for proof. Prop. 3.5 shows that the bounds for both REE (Def. 2.1) and DEE (Def. 2.4) become tighter as $\epsilon$ decreases. Combining Prop. 3.4 and 3.5 implies that minimizing the IPE for general smooth signals reduces both the REE and DEE graph coarsening metrics.

**Complexity.** Our approach formulates coarsening as an optimization problem, so its applicability depends on both computational complexity and convergence time. In App. C, we compare the complexity of our methods with FGC [14], an optimization-based coarsening approach considered

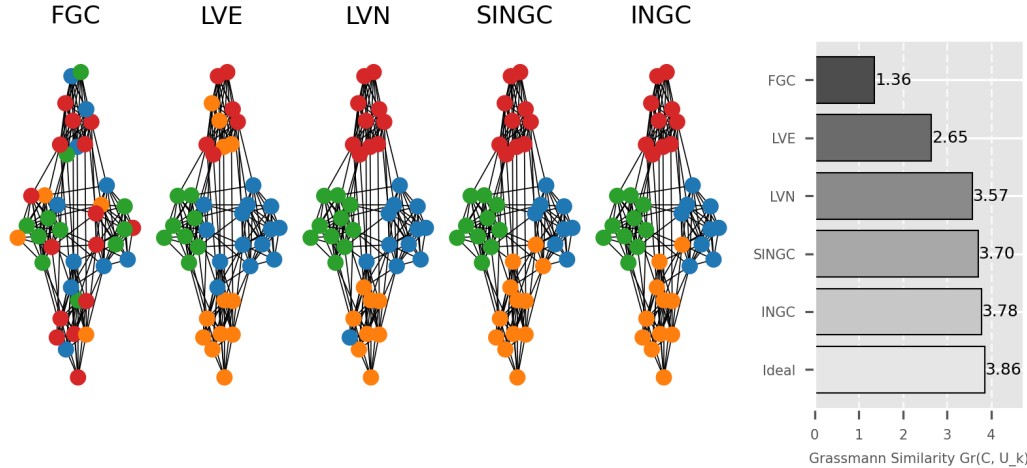

Figure 1: Node assignments of all methods on a synthetic graph generated from an SBM. Nodes with the same color belong to the same class (super-node). The bar plot on the right shows the Grassmann similarity between each method's coarsening matrix $C$ and the leading four eigenvectors $U^{(k)}$ of the graph Laplacian. The bottom bar shows the similarity for the ideal partitioning—i.e., between the block-model-based $C$ and $U^{(k)}$. The maximum similarity in this case is 4.

state-of-the-art on many benchmarks, which has been shown to accelerate GNNs training time. We show that our methods are similarly efficient while achieving better results. Moreover, we show that applying coarsening before a GCN is particularly advantageous for dense graphs with significantly more edges than nodes. Runtime comparisons supporting this claim are presented in App. C. Finally, App. D provides convergence plots illustrating the trade-off between speed and final objective value.

## 4 Experimental results

**Visual illustration.** A key aspect of graph coarsening is how well the global graph's structure is preserved. This can be assessed by how effectively the partitioning into super-nodes captures it. To illustrate this property, we provide a visual example showing how the super-nodes generated by our method align with the graph's global structure. This example highlights how our approach maintains a meaningful graph representation, despite focusing solely on functional relationships.

The example is based on a synthetic graph generated using a Stochastic Block Model (SBM) [30] with four classes, each containing 10 vertices ($N = 40$), an intra-class probability of $p = 0.9$, and an inter-class probability of $q = 0.05$. The feature matrix $X$ is generated following the same graph signal generation mechanism described in (10). We set the target coarsened graph for all coarsening methods to have $k = 4$ super-nodes. In Figure 1, we present the results obtained by our methods (INGC/SINGC), alongside three other graph coarsening baselines: Feature-based Graph Coarsening (FGC) [14], which incorporates node features into the coarsening process, and the Local Variation Neighborhood (LVN) and Local Variation Edges (LVE) methods [10], which use the original graph Laplacian eigenvectors as part of their coarsening objectives. We observe that the super-nodes assignment of our methods closely aligns with the partitioning of the nodes to four classes according to the SBM, as indicated by the node colors in Figure 1. On the right-hand side of Figure 1, we present a bar plot comparing the Grassmann similarity between the coarsening matrices $C$ (which encode the vertex partitioning) produced by each method and the top four eigenvectors of the original graph Laplacian, $U^{(k)}$. The bottom bar represents the matrix $C$ that encodes the ideal partition based on the underlying block model of the graph, serving as a baseline. We observe that our method achieves the highest similarity, closely approaching the ideal partitioning. Note that when the coarsening matrix $C$ and the eigenvector matrix $U^{(k)}$ span the same subspace, the Grassmann similarity reaches its maximum of 4. This comparison highlights our motivation for incorporating Grassmann similarity into our coarsening objective, as it preserves the graph's global structure. Numerically, this property is reflected in the REE metric; we present an extensive evaluation of this and other metrics in the following section. For more visual comparisons, see App.B.

Table 1: Comparison of coarsening methods on four datasets using multiple metrics and coarsening ratios ($r$). For each method, we report REE, RE, HE, DEE, and INP on each dataset. Best results are in bold; second-best are underlined. The last two columns count how often each method achieved the best or second-best performance.

| Method | | Karate Club | | | Les Miserables | | | Cora | | | Citeseer | | | #Best | #2-Best |
|---|---|---|---|---|---|---|---|---|---|---|---|---|---|---|---|
| | $r$ | 0.7 | 0.5 | 0.3 | 0.7 | 0.5 | 0.3 | 0.7 | 0.5 | 0.3 | 0.7 | 0.5 | 0.3 | | |
| LVN | REE | **0.30** | 1.52 | 3.15 | 0.36 | 1.39 | 7.82 | **0.57** | 1.31 | 4.23 | **0.68** | 1.58 | 4.11 | 3 | 4 |
| | RE | 9.71 | 9.87 | 10.31 | 11.42 | 11.91 | 11.91 | 11.42 | 11.62 | 11.70 | 10.85 | 11.05 | 11.14 | 0 | 0 |
| | HE | 1.74 | 1.89 | 2.25 | 1.92 | 2.60 | 2.54 | 1.89 | 2.50 | 3.17 | 1.93 | 2.52 | 3.43 | 0 | 0 |
| | DEE | 36.2 | 47.8 | 46.6 | 65.1 | 99.4 | 90.2 | 39.1 | 70.7 | 90.2 | 40.0 | 66.3 | 111.1 | 0 | 0 |
| | IPE | 0.71 | 0.72 | 1.09 | 2.02 | 2.56 | 1.91 | 2.87 | 2.23 | 1.73 | 0.93 | 0.98 | 1.09 | 0 | 0 |
| LVE | REE | 0.82 | **0.48** | **2.26** | 1.05 | 4.56 | **7.60** | 0.81 | 1.94 | 5.14 | 0.79 | 1.62 | 4.26 | 3 | 2 |
| | RE | 9.39 | 9.91 | 9.97 | 11.55 | 12.14 | 12.48 | 10.62 | 11.51 | 11.69 | 10.52 | 11.02 | 11.14 | 0 | 0 |
| | HE | 1.40 | 1.93 | 2.31 | 1.63 | 2.16 | 2.89 | 1.29 | 2.31 | 3.10 | 1.61 | 2.50 | 3.40 | 0 | 0 |
| | DEE | 17.3 | 39.0 | 84.8 | 19.1 | 20.6 | 63.6 | 22.5 | 44.9 | 78.1 | 30.1 | 63.5 | 109.0 | 0 | 0 |
| | IPE | 0.66 | 0.88 | 0.92 | 1.59 | 2.77 | 3.87 | 0.58 | 1.19 | 1.54 | 0.61 | 0.79 | 0.88 | 0 | 0 |
| FGC | REE | 1.35 | 3.94 | 7.54 | 3.08 | 10.31 | 33.18 | 1.78 | 5.40 | 15.99 | 1.58 | 8.92 | 35.88 | 0 | 0 |
| | RE | 8.70 | 8.81 | 9.26 | 9.97 | 9.98 | 10.91 | 9.70 | 10.82 | 10.76 | 10.47 | 10.23 | 10.31 | 0 | 1 |
| | HE | 1.03 | 1.23 | 1.80 | 0.82 | 0.87 | 1.58 | 0.76 | 1.40 | 1.56 | 1.89 | 1.39 | 1.61 | 0 | 2 |
| | DEE | 8.05 | 11.57 | 21.36 | 4.78 | 7.65 | 9.74 | 0.20 | 0.41 | 5.01 | 19.5 | 7.87 | 1.34 | 0 | 1 |
| | IPE | 0.55 | 0.58 | 0.94 | 1.05 | 2.17 | 3.67 | 0.41 | 1.01 | 3.67 | 1.10 | 0.49 | 0.68 | 0 | 0 |
| INGC (Ours) | REE | 0.78 | 1.30 | 2.97 | **0.08** | **1.30** | 10.40 | 0.86 | **0.84** | **0.76** | 0.71 | **0.62** | **0.42** | 6 | 4 |
| | RE | 6.27 | **7.00** | **8.28** | **5.43** | **8.08** | **9.79** | **9.49** | **10.17** | **10.42** | **8.86** | 9.85 | 10.10 | 9 | 3 |
| | HE | 0.29 | **0.45** | **1.00** | **0.08** | **0.33** | **0.83** | **0.67** | **1.04** | **1.37** | **0.64** | 1.21 | 1.61 | 9 | 3 |
| | DEE | **0.01** | **0.03** | **0.02** | **0.04** | **0.02** | **0.10** | **0.03** | **0.40** | **3.12** | **0.02** | **0.66** | 1.01 | 11 | 1 |
| | IPE | 0.19 | **0.31** | **0.48** | 0.30 | **0.57** | **0.86** | **0.31** | **0.43** | **0.68** | **0.24** | 0.34 | 0.58 | 8 | 4 |
| SINGC (Ours) | REE | 0.86 | 1.78 | 4.02 | 0.50 | 2.32 | 7.78 | 0.86 | **0.84** | 5.10 | 0.83 | **0.62** | **0.42** | 3 | 0 |
| | RE | **6.09** | 8.05 | 8.74 | 9.07 | 9.60 | 10.49 | 9.54 | **10.17** | 10.95 | 9.32 | **9.76** | **9.92** | 4 | 7 |
| | HE | **0.27** | 0.85 | 1.46 | 0.51 | 0.72 | 1.29 | 0.69 | **1.04** | 1.84 | 0.83 | **1.14** | **1.27** | 4 | 7 |
| | DEE | 0.02 | 0.51 | 6.56 | 0.47 | 0.35 | **0.10** | **0.03** | **0.40** | 8.12 | 1.01 | 1.45 | **0.10** | 4 | 7 |
| | IPE | **0.19** | 0.43 | 0.59 | **0.21** | 0.63 | 1.07 | **0.31** | **0.43** | 0.69 | 0.27 | **0.25** | **0.47** | 6 | 6 |

**Graph coarsening metrics.**  Here, we evaluate the performance of our methods on several benchmark datasets using the coarsening metrics from Sec. 2. We compare them with current SOTA coarsening methods—LVN and LVE—which are known to perform best on structural-preservation metrics (e.g., REE, RE), and FGC, which is considered the leading method for metrics that also consider the features of the nodes. We conduct experiments on four datasets: The Karate Club[31], Les Miserables[32], Cora[33], and Citeseer[34]. Note that the Cora and Citeseer datasets include node features, whereas the Karate Club and Les Miserables datasets do not. For the latter two, we generated node features using the signal generation mechanism presented in Section 3.1.

Table 1 summarizes the performance of our methods (INGC and SINGC) and the baselines (FGC, LVN and LVE) across different datasets and coarsening ratios ($r = \frac{k}{n} = 0.7$, 0.5, and 0.3). The best performance for each metric is highlighted in bold, and the second-best is underlined. The last two columns summarize the number of settings in which each method achieved the lowest or second-lowest score compared to others. We observe that INGC achieves the best overall performance across all graph metrics. SINGC, a more efficient variant, is also highly competitive—often ranking second and occasionally achieving the best results. Baseline methods (LVN, LVE, and FGC) show mixed performance. LVN and LVE perform well on REE for some datasets, indicating good spectral preservation, but generally underperform on metrics involving node features. FGC, the only baseline that incorporates node features in coarsening, outperforms the others on related metrics. However, both our methods consistently surpass FGC. The strong RE score of our approach demonstrates its broader ability to preserve graph structure, even beyond the theoretical setting of Theorem 3.2, as the datasets used are connected or have far fewer than $n - k$ connected components.

The hyperparameters in these experiments were selected via grid search, a standard approach in graph coarsening when aiming to minimize a specific metric. The best results for each metric were obtained using different hyperparameters, as reported in App. G.3. This variability highlights the

Table 2: Node classification accuracy across datasets and coarsening ratios ($r$). Best results are bolded; second-best are underlined. The last two rows count best and second-best results per method.

| Dataset | r | GCOND | SCAL(LV) | FGC | MGC | INGC (Ours) | SINGC (Ours) |
|---------|------|-------|----------|-----|-----|-------------|--------------|
| Cora | 0.3 | $81.56 \pm 0.60$ | $79.42 \pm 1.71$ | $\underline{85.79 \pm 0.24}$ | $84.56 \pm 1.40$ | $\mathbf{87.55 \pm 0.16}$ | $84.51 \pm 0.33$ |
| | 0.1 | $81.37 \pm 0.40$ | $71.38 \pm 3.62$ | $\underline{81.46 \pm 0.79}$ | $76.02 \pm 0.93$ | $\mathbf{83.38 \pm 0.47}$ | $82.76 \pm 0.32$ |
| | 0.05 | $79.93 \pm 0.44$ | $55.32 \pm 7.03$ | $\mathbf{80.01 \pm 0.51}$ | - | $77.42 \pm 0.78$ | $77.81 \pm 0.68$ |
| Citeseer | 0.3 | $72.43 \pm 0.94$ | $68.87 \pm 1.37$ | $74.64 \pm 1.37$ | $74.60 \pm 1.20$ | $\mathbf{76.89 \pm 0.23}$ | $\underline{76.66 \pm 0.27}$ |
| | 0.1 | $70.46 \pm 0.47$ | $71.38 \pm 3.62$ | $\mathbf{73.36 \pm 0.53}$ | $70.57 \pm 1.25$ | $\underline{72.63 \pm 0.25}$ | $69.71 \pm 0.72$ |
| | 0.05 | $64.03 \pm 2.40$ | $55.32 \pm 7.03$ | $\mathbf{71.02 \pm 0.96}$ | - | $66.02 \pm 0.32$ | $\underline{66.37 \pm 0.57}$ |
| Co-phy | 0.05 | $93.05 \pm 0.26$ | $73.09 \pm 7.41$ | $94.27 \pm 0.25$ | $\mathbf{94.52 \pm 0.19}$ | $\underline{94.29 \pm 0.10}$ | $94.04 \pm 0.06$ |
| | 0.03 | $92.81 \pm 0.31$ | $63.65 \pm 9.65$ | $\underline{94.02 \pm 0.20}$ | $93.64 \pm 0.25$ | $\mathbf{94.20 \pm 0.13}$ | $93.52 \pm 0.13$ |
| | 0.01 | $92.79 \pm 0.40$ | $31.08 \pm 2.65$ | $93.08 \pm 0.22$ | - | $\mathbf{93.95 \pm 0.20}$ | $93.20 \pm 0.10$ |
| Pubmed | 0.05 | $78.16 \pm 0.30$ | $72.82 \pm 2.62$ | $80.73 \pm 0.44$ | $81.89 \pm 0.01$ | $\mathbf{83.59 \pm 0.22}$ | $\underline{83.55 \pm 0.32}$ |
| | 0.03 | $78.04 \pm 0.47$ | $70.24 \pm 2.63$ | $79.91 \pm 0.30$ | $80.70 \pm 0.01$ | $81.93 \pm 0.22$ | $\mathbf{83.19 \pm 0.18}$ |
| | 0.01 | $77.20 \pm 0.20$ | $54.49 \pm 10.5$ | $78.42 \pm 0.43$ | - | $\underline{79.09 \pm 0.26}$ | $\mathbf{79.96 \pm 0.34}$ |
| Co-CS | 0.05 | $86.29 \pm 0.63$ | $34.45 \pm 10.0$ | $89.60 \pm 0.39$ | - | $\underline{90.84 \pm 0.12}$ | $\mathbf{90.92 \pm 0.22}$ |
| | 0.03 | $86.32 \pm 0.45$ | $26.06 \pm 9.29$ | $88.29 \pm 0.79$ | - | $\underline{89.59 \pm 0.38}$ | $\mathbf{89.99 \pm 0.41}$ |
| | 0.01 | $84.01 \pm 0.02$ | $14.42 \pm 8.50$ | $\underline{86.37 \pm 1.36}$ | - | $\mathbf{87.93 \pm 0.33}$ | $83.39 \pm 0.33$ |
| #Best | | 0 | 0 | 3 | 1 | 7 | 4 |
| #2-Best | | 1 | 0 | 3 | 1 | 5 | 4 |

need to tune hyperparameters based on the specific application and the most relevant coarsening metric. For example, minimizing REE may be crucial for clustering, while tasks like graph pooling and node classification—which rely heavily on node features—benefit from prioritizing DEE and INP during tuning. App. F presents a hyperparameter study showing each parameter's contribution and sensitivity across metrics.

**Node classification.** Next, we evaluate our method on node classification using several benchmark datasets. This task assesses how well coarsened graphs preserve structural and feature information for accurate label prediction [4]. Following Kumar et al. [14], we train a Graph Neural Network (GNN) on the coarsened graph and predict node labels for the original graph, reducing training time due to fewer nodes and edges. Specifically, we: (1) generate a coarsened graph using a selected method; (2) compute super-node labels via $\boldsymbol{y}_c = \boldsymbol{C}^\dagger \boldsymbol{y}$; (3) train a GNN on the coarsened graph; and (4) evaluate predictions $\hat{\boldsymbol{y}} = \text{GNN}(\boldsymbol{L}, \boldsymbol{X})$ against the original labels $\boldsymbol{y}$. Note that this task is used solely for evaluation, with all labels $\boldsymbol{y}$ available throughout.

We replicated the experimental settings from Kumar et al. [14] and Huang et al. [35], both of which employ a Graph Convolutional Network (GCN) [36]. We then compared the performance of our methods (INGC and SINGC) with the top-performing methods in those papers that are considered to be SOTA, namely SCAL [35], Featured Graph Coarsening (FGC) [14], Graph Condensation (GCOND) [7], and Multi-Component Coarsening (MGC) [37]. The datasets include two medium-sized graphs (Cora and Citeseer) and three large-scale graphs (Co-Physics, Pubmed, and Co-CS). Each method's effectiveness was evaluated using 10-fold cross-validation. A detailed description of the experimental setup is in App. G.2.

The results are reported in Table 2 with mean accuracy and standard deviation across the folds for each method at different coarsening ratios $r$. We observe that INGC outperforms the SOTA methods on large datasets (Co-Physics, Pubmed, and Co-CS), and matches their performance on medium-sized datasets. SINGC outperforms baseline methods in most settings, despite having a simpler optimization objective. A key takeaway from the results is that the integration of node features into the coarsening process gives FGC, INGC and SINGC a competitive advantage over methods that primarily focus on structural properties, such as SCAL and GCOND, as was also indicated in [14]. This relationship is intuitive, as node classification relies not only on structural relationships but also on the meaningful preservation of node features.

App. F presents an ablation study on $\beta$, highlighting the role of the Grassmann similarity term. App. E discusses Assumption 3.3, evaluates its validity on real datasets, and shows our method remains effective even when it is not fully met. Moreover, our empirical results indicate that SINGC benefits when a larger fraction of signal energy resides in the low-frequency band, and its best performance reported on PubMed, where this assumption is most valid.

The source code for all experiments is available at: `Code`.

# 5 Conclusion

In this paper, we introduced a new graph coarsening method that focuses on preserving the inner products of graph signals during the coarsening process. We demonstrated that, although primarily considering node features, our approach also maintains the global structure of the graph. Our methods, INGC and SINGC, outperform state-of-the-art techniques across various graph coarsening metrics and tasks (e.g., node classification), showcasing their versatility and effectiveness in preserving essential graph properties. These results highlight the potential of our approach for graph-based learning applications. However, similar to other optimization-based coarsening techniques, our method faces scalability limitations when applied to extremely large graphs. This challenge may be addressed in practical scenarios by partitioning the graph into manageable subgraphs, processing each subgraph independently, potentially in parallel, and subsequently reconnecting them [38]. A critical assumption underlying our work is that the node features exhibit smoothness with respect to the graph topology, an attribute closely related to homophily. Consequently, our evaluation is restricted to homophilic graphs. For future research, an extension to heterophilic datasets could involve adapting the second term in our objective to facilitate alignment with alternative frequency bands. For example, the low-frequency basis $U^{(k)}$ may be replaced with a different basis representing mid or high frequencies, guided by prior knowledge of label–frequency alignment. Future directions also include integrating our coarsening into graph pooling and evaluating its impact on improving GNN performance.

## Acknowledgments

We thank the anonymous reviewers for their insightful feedback. This work was supported by the European Union's Horizon 2020 research and innovation programme under grant agreement No. 802735-ERC-DIFFOP.

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

# A  Appendix - Theorems' proofs

## A.1  Proof of Proposition 3.2

[Proof of Proposition 3.2] Given a graph $\mathcal{G}$ with a graph Laplacian $\boldsymbol{L}$ and its coarsened graph $\mathcal{G}_c$ with a graph Laplacian $\boldsymbol{L}_c = \boldsymbol{C}^T \boldsymbol{L} \boldsymbol{C}$, and assume that for any two graph signals $\boldsymbol{x}, \boldsymbol{y} \in \mathbb{R}^n$, the following condition is satisfied:

$$\boldsymbol{x}^\top \boldsymbol{L} \boldsymbol{y} = \boldsymbol{x}_c^\top \boldsymbol{L}_c \boldsymbol{y}_c \tag{11}$$

We plug in the the definitions of $\boldsymbol{x}_c = \boldsymbol{P}\boldsymbol{x}$ and $\boldsymbol{y}_c = \boldsymbol{P}\boldsymbol{y}$ in (11) and obtain:

$$\begin{aligned}
\boldsymbol{x}^\top \boldsymbol{L} \boldsymbol{y} &= \boldsymbol{x}_c^\top \boldsymbol{L}_c \boldsymbol{y}_c \\
&= (\boldsymbol{P}\boldsymbol{x})^\top \boldsymbol{L}_c \boldsymbol{P}\boldsymbol{y} \\
&= \boldsymbol{x}^\top \boldsymbol{P}^\top \boldsymbol{L}_c \boldsymbol{P}\boldsymbol{y} \\
&= \boldsymbol{x}^\top \boldsymbol{L}_l \boldsymbol{y}.
\end{aligned} \tag{12}$$

where in the last equality we plug-in the definition of the lifted Laplacian (reconstructed Laplacian) $\boldsymbol{L}_l = \boldsymbol{P}^T \boldsymbol{L}_c \boldsymbol{P}$.

Assuming (12) holds for every pair of signals $\boldsymbol{x}, \boldsymbol{y} \in \mathbb{R}^n$, one can choose specific signals such that $\boldsymbol{L}[i, j] = \boldsymbol{L}_l[i, j]$ for all $i, j = 1, \dots, n$, allowing us to conclude:

$$\boldsymbol{L} = \boldsymbol{L}_l$$

This implies that the full Laplacian $\boldsymbol{L}$ can be fully reconstructed for $\boldsymbol{L}_c$.

## A.2  Proof of Proposition 3.4

[Proof of Proposition 3.4] Let $\boldsymbol{x}, \boldsymbol{y} \in \mathbb{R}^n$ be two k-smooth signals on the graph $\mathcal{G}$ with graph Laplacian $\boldsymbol{L}$. Define $\boldsymbol{x}_c = \boldsymbol{C}^\top \boldsymbol{x}, \boldsymbol{y}_c = \boldsymbol{C}^\top \boldsymbol{y} \in \mathbb{R}^k$, and let $\boldsymbol{L}_c = \boldsymbol{C}^\top \boldsymbol{L} \boldsymbol{C}$, where $\boldsymbol{C} = \boldsymbol{U}^{(k)}\boldsymbol{O}$, and $\boldsymbol{O} \in \mathcal{O}$ is some k-dimension rotation matrix. Then, the following relation holds:

$$\begin{aligned}
\boldsymbol{x}^\top \boldsymbol{L} \boldsymbol{y} - \boldsymbol{x}_c^\top \boldsymbol{L}_c \boldsymbol{y}_c &= \boldsymbol{x}^\top \boldsymbol{L} \boldsymbol{y} - ((\boldsymbol{C}^\top \boldsymbol{x})^\top \boldsymbol{C}^\top \boldsymbol{L} \boldsymbol{C} \boldsymbol{C}^\top \boldsymbol{y} \\
&= \boldsymbol{x}^\top \boldsymbol{L} \boldsymbol{y} - ((\boldsymbol{U}^{(k)}\boldsymbol{O})^\top \boldsymbol{x})^\top (\boldsymbol{U}^{(k)}\boldsymbol{O})^\top \boldsymbol{L} \boldsymbol{U}^{(k)}\boldsymbol{O}(\boldsymbol{U}^{(k)}\boldsymbol{O})^\top \boldsymbol{y} \\
&= \boldsymbol{x}^\top \boldsymbol{L} \boldsymbol{y} - \boldsymbol{x}^\top \boldsymbol{U}^{(k)}\boldsymbol{O}\boldsymbol{O}^\top(\boldsymbol{U}^{(k)})^\top \boldsymbol{L} \boldsymbol{U}^{(k)}\boldsymbol{O}\boldsymbol{O}^\top(\boldsymbol{U}^{(k)})^\top \boldsymbol{y} \\
&= \boldsymbol{x}^\top \boldsymbol{L} \boldsymbol{y} - \boldsymbol{x}^\top \boldsymbol{U}^{(k)}(\boldsymbol{U}^{(k)})^\top \boldsymbol{L} \boldsymbol{U}^{(k)}(\boldsymbol{U}^{(k)})^\top \boldsymbol{y} \\
&= \boldsymbol{x}^\top \boldsymbol{L} \boldsymbol{y} - \boldsymbol{x}^\top \boldsymbol{L} \boldsymbol{y} = 0
\end{aligned}$$

where the third equality holds because $\boldsymbol{O}$ is a rotation matrix satisfying $\boldsymbol{O}\boldsymbol{O}^\top = \boldsymbol{I}_{k \times k}$. The fifth equality holds because $\boldsymbol{x}$ and $\boldsymbol{y}$ are k-smooth and satisfy $\boldsymbol{x} = \boldsymbol{U}^{(k)}(\boldsymbol{U}^{(k)})^\top \boldsymbol{x}$ and $\boldsymbol{y} = \boldsymbol{U}^{(k)}(\boldsymbol{U}^{(k)})^\top \boldsymbol{y}$.

Thus, for any matrix $\boldsymbol{X}$ whose columns are k-smooth signals, we have:

$$\|\boldsymbol{X}^\top \boldsymbol{L} \boldsymbol{X} - \boldsymbol{X}_c^\top \boldsymbol{L}_c \boldsymbol{X}_c\|_F^2 = 0.$$

## A.3  Proof of Proposition 3.5

The proof of Proposition 3.5 relies on the following definition and lemma.

**Definition A.1 (Restricted spectral approximation [10])** *Let $R$ be a k-dimensional subspace of $\mathbb{R}^n$. Matrices $\boldsymbol{L}_c$ and $\boldsymbol{L}$ are $(R, \epsilon)$-similar if there exists an $\epsilon > 0$ such that*

$$\|\boldsymbol{x} - \boldsymbol{x}_l\|_L \leq \epsilon \|\boldsymbol{x}\|_L, \quad \text{for all } \boldsymbol{x} \in R,$$

*where $\boldsymbol{x}_l = \boldsymbol{C}\boldsymbol{C}^\dagger \boldsymbol{x}$.*

**Lemma A.2** *Let $\boldsymbol{L}$ be the Laplacian matrix of a connected graph $\mathcal{G}$, and let $\boldsymbol{U}^{(k)}$ be the matrix containing its $k$-leading eigenvectors. Suppose $\boldsymbol{L}_c = \boldsymbol{C}^\top \boldsymbol{L} \boldsymbol{C}$ is the Laplacian matrix of a coarsened graph derived using a coarsening operator $\boldsymbol{C}$ with normalized columns such that*

$$tr(\boldsymbol{U}^{(k)}(\boldsymbol{U}^{(k)})^\top \boldsymbol{C}\boldsymbol{C}^\top) = k - \epsilon.$$

*Then, the matrices $\boldsymbol{L}$ and $\boldsymbol{L}_c$ are $(R, \epsilon\kappa)$-similar, where $\kappa = \frac{\lambda_{max}(\boldsymbol{L})}{\lambda_2(\boldsymbol{L})}$, and $R = span(\boldsymbol{U}^{(k)})$*

[Proof of Lemma A.2] We note the projection matrix defined by $\boldsymbol{C}$ as $\boldsymbol{\Pi}_C = \boldsymbol{C}\boldsymbol{C}^\top$. Given the trace condition $\text{tr}(\boldsymbol{U}^{(k)}(\boldsymbol{U}^{(k)})^\top \boldsymbol{\Pi}_C) = k - \epsilon$, we can express it as:

$$\text{tr}(\boldsymbol{U}^{(k)}(\boldsymbol{U}^{(k)})^\top \boldsymbol{\Pi}_C) = \text{tr}((\boldsymbol{U}^{(k)})^\top \boldsymbol{\Pi}_C \boldsymbol{U}^{(k)}) = k - \epsilon.$$

From this, we immediately obtain:

$$\text{tr}((\boldsymbol{U}^{(k)})^\top (\boldsymbol{I} - \boldsymbol{\Pi}_C)\boldsymbol{U}^{(k)}) = \epsilon. \tag{13}$$

This follows from the fact that:

$$k = \text{tr}((\boldsymbol{U}^{(k)})^\top \boldsymbol{U}^{(k)}) = \text{tr}(\boldsymbol{U}^{(k)}(\boldsymbol{U}^{(k)})^\top) = \text{tr}(\boldsymbol{U}^{(k)}\boldsymbol{I}_{n\times n}\boldsymbol{U}^{(k)})$$
$$= \text{tr}((\boldsymbol{U}^{(k)})^\top \boldsymbol{\Pi}_C \boldsymbol{U}^{(k)}) + \text{tr}((\boldsymbol{U}^{(k)})^\top (\boldsymbol{I} - \boldsymbol{\Pi}_C)\boldsymbol{U}^{(k)}),$$

where the first equality holds because $\boldsymbol{U}^{(k)}$ is orthonormal matrix.

Next, we express the term $\|\boldsymbol{x} - \boldsymbol{x}_l\|_L$. Since the columns of $\boldsymbol{C}$ are orthogonal, we can use $\boldsymbol{x}_l = \boldsymbol{C}\boldsymbol{P}\boldsymbol{x} = \boldsymbol{C}\boldsymbol{C}^\dagger \boldsymbol{x} = \boldsymbol{C}\boldsymbol{C}^\top \boldsymbol{x}$, and obtain:

$$\|\boldsymbol{x} - \boldsymbol{x}_l\|_L = (\boldsymbol{x} - \boldsymbol{C}\boldsymbol{C}^\top \boldsymbol{x})^\top \boldsymbol{L}(\boldsymbol{x} - \boldsymbol{C}\boldsymbol{C}^\top \boldsymbol{x})$$
$$= ((\boldsymbol{I} - \boldsymbol{C}\boldsymbol{C}^\top)\boldsymbol{x})^\top \boldsymbol{L}(\boldsymbol{I} - \boldsymbol{C}\boldsymbol{C}^\top)\boldsymbol{x}. \tag{14}$$

Using the Rayleigh quotient [39], we can bound by:

$$\|\boldsymbol{x} - \boldsymbol{x}_l\|_L = ((\boldsymbol{I} - \boldsymbol{C}\boldsymbol{C}^\top)\boldsymbol{x})^\top \boldsymbol{L}((\boldsymbol{I} - \boldsymbol{C}\boldsymbol{C}^\top)\boldsymbol{x})$$
$$\leq \lambda_{\max}(\boldsymbol{L})\|(\boldsymbol{I} - \boldsymbol{C}\boldsymbol{C}^\top)\boldsymbol{x}\|_2^2. \tag{15}$$

Next, we proceed to bound the term $\|(\boldsymbol{I} - \boldsymbol{C}\boldsymbol{C}^\top)\boldsymbol{x}\|_2^2$. Since $\boldsymbol{x}$ is spanned by $\boldsymbol{U}^{(k)}$, we write $\boldsymbol{x} = \boldsymbol{U}^{(k)}\boldsymbol{z}$. Therefore, we get:

$$\|(\boldsymbol{I} - \boldsymbol{C}\boldsymbol{C}^\top)\boldsymbol{x}\|_2^2 = \boldsymbol{z}^\top (\boldsymbol{U}^{(k)})^\top (\boldsymbol{I} - \boldsymbol{C}\boldsymbol{C}^\top)\boldsymbol{U}^{(k)}\boldsymbol{z}.$$

From (13), we know that the maximum eigenvalue of $(\boldsymbol{U}^{(k)})^\top (\boldsymbol{I} - \boldsymbol{\Pi}_C)\boldsymbol{U}^{(k)}$ is bounded by $\epsilon$. Thus, by applying the Rayleigh quotient, we obtain:

$$\|(\boldsymbol{I} - \boldsymbol{C}\boldsymbol{C}^\top)\boldsymbol{x}\|_2^2 \leq \epsilon\|\boldsymbol{z}\|_2^2 = \epsilon\|\boldsymbol{x}\|_2^2.$$

Substituting this bound into (15), we have:

$$\|\boldsymbol{x} - \boldsymbol{x}_l\|_L \leq \epsilon\lambda_{\max}(\boldsymbol{L})\|\boldsymbol{x}\|_2^2.$$

Since $\boldsymbol{L}$ is the graph Laplacian of a connected graph, it has only one zero eigenvalue, corresponding to the constant vector. Assuming $\boldsymbol{x}$ is not a constant vector, we can bound $\|\boldsymbol{x}\|_2^2$ using the Rayleigh quotient:

$$\|\boldsymbol{x}\|_2^2 \geq \frac{\|\boldsymbol{x}\|_L}{\lambda_2(\boldsymbol{L})}.$$

Substituting this into the previous inequality, we obtain:

$$\|\boldsymbol{x} - \boldsymbol{x}_l\|_L \leq \epsilon \frac{\lambda_{\max}(\boldsymbol{L})}{\lambda_2(\boldsymbol{L})} \|\boldsymbol{x}\|_L = \epsilon\kappa\|\boldsymbol{x}\|_L,$$

where $\kappa = \frac{\lambda_{\max}(\boldsymbol{L})}{\lambda_2(\boldsymbol{L})}$ is the condition number of $\boldsymbol{L}$.

Finally, if $\boldsymbol{x}$ is a constant vector, then since the columns of $\boldsymbol{C}$ span the constant vector, we have:

$$\|\boldsymbol{x} - \boldsymbol{x}_l\|_L = \|\boldsymbol{x} - \boldsymbol{C}\boldsymbol{C}^\top\boldsymbol{x}\|_L = 0.$$

Thus, for all $\boldsymbol{x} \in \operatorname{span}(\boldsymbol{U}^{(k)})$, we conclude that:

$$\|\boldsymbol{x} - \boldsymbol{x}_l\|_L \leq \epsilon\kappa\|\boldsymbol{x}\|_L.$$

i.e $\boldsymbol{L}$ and $\boldsymbol{L}_c$ are $(R, \epsilon\kappa)$ similar.

Then, according to Theorem 13 and Corollary 12 in [20], if the full graph Laplacian $\boldsymbol{L}$ and the coarsen graph Laplacian are $\boldsymbol{L}_c$ are $(\boldsymbol{U}^{(k)}, \epsilon\kappa)$-similar, they satisfy the following inequalities: :

$$(1 - \epsilon\kappa)\|\boldsymbol{x}\|_L \leq \|\boldsymbol{x}\|_{L_c} \leq (1 + \epsilon\kappa)\|\boldsymbol{x}\|_L$$

$$\frac{1}{\mu_1}\lambda^{(i)} \leq \lambda_c^{(i)} \leq \frac{1}{\mu_2} \frac{(1 + \epsilon\kappa)^2}{1 - (\epsilon\kappa)^2(\lambda^{(i)}/\lambda^{(2)})}\lambda^{(i)}, \qquad 2 \leq i \leq k$$

according to Proposition 3.5, where $\kappa = \frac{\lambda_{\max}(\boldsymbol{L})}{\lambda_2(\boldsymbol{L})}$, and $\mu_1, \mu_2$ and the first and $k$ eigenvalues of the matrix $\boldsymbol{P}\boldsymbol{P}^\top$.

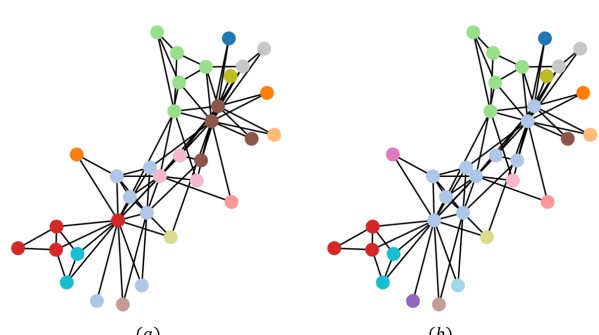

(a)  (b)

Figure 2: Clustering of the Karate Club network using (a) our SINGC method and (b) conventional $k$-means clustering on the leading eigenvectors. Each color represents a distinct cluster, and the coarsened graph is obtained by mapping nodes from the same cluster (color) to a single super-node. We observe that the clusters produced by SINGC are more balanced, which is advantageous for downstream graph learning tasks.

## A.4  Gradient Computation

We start with a recap of our suggested objective function:

$$\min_{\boldsymbol{X}_c, \boldsymbol{C}} f(\boldsymbol{X}_c, \boldsymbol{C}) = \|\boldsymbol{X}^\top \boldsymbol{L}\boldsymbol{X} - \boldsymbol{X}_c^\top \boldsymbol{C}^\top \boldsymbol{L}\boldsymbol{C}\boldsymbol{X}_c\|_F - \beta tr(\boldsymbol{U}^{(k)}(\boldsymbol{U}^{(k)})^\top \boldsymbol{C}\boldsymbol{C}^\top)$$

$$+ \lambda\|\boldsymbol{C}^T\|_{1,2}^2 - \alpha \log\det(\boldsymbol{L}_c + \boldsymbol{J})$$

$$\textbf{s.t.} \qquad \boldsymbol{X}_c = \boldsymbol{C}^\dagger \boldsymbol{X}$$

The gradient of each term with respect to $C$ is:

$$\nabla_C(-tr(U^{(k)}(U^{(k)})^\top CC^\top)) = -2U^{(k)}(U^{(k)})^\top C$$

$$\nabla_C(\|X^\top LX - X_c^\top C^\top LCX_c\|_F) = -(2LCX_c(X^\top LX - (LCX_c)^\top CX_c)X_c^\top$$
$$+ 2L^\top CX_c(X^\top LX - (CX_c)^\top LCX_c)X_c^\top)$$

$$\nabla_C(\|C^T\|_{1,2}^2) = C1_{k\times k}$$

$$\nabla_C(logdet(L_c + J)) = LC(C^\top LC + J)^{-1}$$

where the third was shown in [14], assuming all elements of $C$ to non-negative (since $C \in \mathcal{C}$. The full gradient with respect to $C$ of (9) is:

$$\nabla_C f(C, X_c) = 2\beta U^{(k)}(U^{(k)})^\top C + \lambda C$$
$$- (2LCX_c(X^\top LX - (LCX_c)^\top CX_c)X_c^\top$$
$$+ 2L^\top CX_c(X^\top LX - (CX_c)^\top LCX_c)X_c^\top)$$
$$+ \lambda C1_{k\times k} - \alpha(LC(C^\top LC + J)^{-1}) \tag{16}$$

We note that in case of the SINGC Algorithm the computed gradient is simpler and can be express as:

$$\nabla_C \tilde{f}(C, X_c) = 2U^{(k)}(U^{(k)})^\top C + \lambda C1_{k\times k} - \alpha(LC(C^\top LC + J)^{-1}) \tag{17}$$

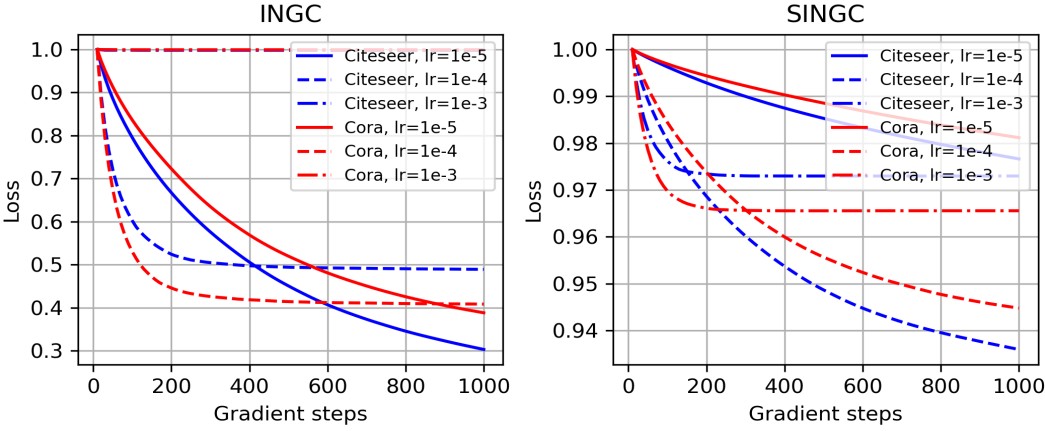

Figure 3: Convergence rates of INGC and SINGC methods for Cora and Citeseer datasets. Citeseer results are shown in blue with varying line styles for different learning rates, while Cora results are shown in red. Gradient steps are on the x-axis, and normalized loss values are on the y-axis.

## B  Methods Performance - Visual Comparison

Table 3: Comparison of gradient expressions and time complexities for FGC, INGC, and SINGC.

| | FGC | INGC | SINGC |
|---|---|---|---|
| **Gradient Expression** | $\nabla_C f(C, X_c) = 2((CX_c - X)$ $+ L(CX_c))X_c^\top$ $+ \lambda C1_{k\times k}$ $- \alpha(LC(C^\top LC + J)^{-1})$ | $\nabla_C f(C, X_c) = 2\beta U^{(k)}(U^{(k)})^\top C$ $- [2L(CX_c)(X^\top LX$ $- (LCX_c)^\top(CX_c))X_c^\top]$ $+ \lambda C1_{k\times k}$ $- \alpha(LC(C^\top LC + J)^{-1})$ | $\nabla_C f(C) = 2U^{(k)}(U^{(k)})^\top C$ $+ \lambda C1_{k\times k}$ $- \alpha(LC(C^\top LC + J)^{-1})$ |
| **Theoretical Time Complexity** | $O(n^2(k+d) + k^3)$ | $O(n^2(k+d)$ $+ ndk + nk^2 + k^3)$ | $O(n^2k + nk^2 + k^3)$ |

In Section 4, we demonstrated the global preservation property of our method on a specific task with a low number of super-nodes, similar to a clustering task. However, in typical coarsening scenarios,

Table 4: Runtime comparison of our methods and FGC for a coarsening ratio of $r = 0.03$, where all times $\tau$ are reported in seconds. Columns 2–5 show the runtime of each coarsening algorithm. Column 6 reports the training time of a GCN on the original graph, while Column 7 reports the training time on the coarsened graph. The final column shows in percentage how much training time was saved by applying coarsening before training, compared to training on the full graph.

| Dataset | FGC | INGC | SINGC | INGC($\beta = 0$) | GCN full | GCN coarse | Speedup |
|---------|-----|------|-------|-------------------|----------|------------|---------|
| Citeseer | 04.67 | 09.38 | 04.04 | 06.11 | 42.02 | 26.03 | 14-29% |
| Pubmed | 20.54 | 75.27 | 49.47 | 33.05 | 93.15 | 23.92 | 21-38% |
| Co-CS | 70.80 | 284.13 | 180.33 | 120.66 | 369.10 | 40.86 | 12-56% |
| Co-Phy | 266.83 | 1200.76 | 720.22 | 540.50 | 883.96 | 52.26 | 12-32% |

there are usually a larger number of super-nodes. In this section, we present the performance of this property in more practical coarsening scenarios, showing how our method continues to preserve the global structure of the graph.

In Figures 5 and 7 , we present the results obtained by our methods (INGC/SINGC), alongside the three baseline methods described in Section 4 on the Karate Club and Les Miserables datasets. Each row in the figures shows the partitioning produced by each method at different coarsening ratios. Nodes of the same color are grouped into the same super-node. We observe that our method groups adjacent nodes into super-nodes, thereby preserving the global structure of the graph.

Since our method leverages the graph Laplacian eigenvectors to partition the vertices, we also compare it to the commonly used spectral clustering approach [19], which partitions the vertices by applying k-means [40] on the leading graph Laplacian eigenvectors. In Figure 2, we present the clustering results obtained by the SINGC method on the well-known Karate Club dataset[31], which consists of $n = 34$ nodes. We apply our method with a target of $k = \frac{n}{2} = 17$ super-nodes and compare the results to those obtained by spectral clustering[19] applied to the top $k$ leading eigenvectors. In the figure, each color represents a distinct cluster.We observe that the clusters produced by our method are more balanced compared to those generated by spectral clustering, which tends to form one large cluster alongside several smaller, single-node clusters. This balance is advantageous for downstream graph learning tasks, such as graph pooling.

## C  Complexity Analysis

For an input graph with $n$ nodes, $e$ edges, and node features of dimension $p$, the coarsened graph has $k$ nodes and $e_c$ edges. The dominant computational cost of the coarsening process arises from the gradient computation performed at each optimization step. Table 3 summarizes the gradient expressions and their time complexities, highlighting that SINGC is the most efficient, while INGC remains competitive with FGC, considering the gradient computation.

Both INGC and SINGC have additional overhead beyond the optimization itself due to the computation of the top-$k$ Laplacian eigenvectors, which typically requires $\mathcal{O}(n^2 k)$ operations. This step can significantly increase the total runtime. However, this cost can be avoided by setting $\beta = 0$, which removes the Grassmann similarity term from the objective. As shown in Table 6, even without this term, our methods still outperform the baselines.

Table 4 reports the runtime of our methods and FGC, demonstrating their efficiency in reducing GNN training time. Specifically, we compare the time required to train a GCN on the original graph versus the coarse graph, including the coarsening time. The results confirm that applying coarsening before GCN training yields substantial time savings as summarized in the last column of the Table.

All runtime values are reported in seconds. Experiments were conducted on a machine with a 12th-Gen Intel i7 CPU, an NVIDIA RTX A5000 GPU, and 64 GB of RAM.

Table 5: Average fraction of signal energy captured by the first $k$ Laplacian eigenvectors (i.e., $k$-smoothness) for each dataset and coarsening ratio.

| Dataset | r=1 | r=0.7 | r=0.5 | r=0.3 | r=0.1 | r=0.05 | r=0.03 | r=0.01 |
|---------|-----|-------|-------|-------|-------|--------|--------|--------|
| Cora | 1 | 0.85 | 0.74 | 0.60 | 0.38 | 0.27 | 0.19 | 0.08 |
| Citeseer | 1 | 0.86 | 0.76 | 0.63 | 0.34 | 0.25 | 0.18 | 0.10 |
| Pubmed | 1 | 0.87 | 0.77 | 0.65 | 0.50 | 0.43 | 0.39 | 0.32 |
| Co-CS | 1 | 0.77 | 0.67 | 0.55 | 0.34 | 0.26 | 0.22 | 0.14 |

## D  Convergence Analysis

Figure 3 illustrates the convergence rates of the INGC and SINGC methods on the Cora dataset for a coarsening ratio $r = 0.3$. The left subplot shows the performance of INGC, while the right subplot depicts SINGC. For both methods, Citeseer results are in blue with varying line styles for different learning rates, and Cora results in red with corresponding line styles. The x-axis represents gradient steps, and the y-axis shows normalized loss values.

The results reveal a typical convergence pattern for different learning rates. A trade-off is observed between convergence speed and final objective loss: higher learning rates lead to faster convergence but result in a higher final loss. This phenomenon is consistent across both datasets. We note that recent work has shown that lower learning rates can achieve lower minimal loss values but may risk unstable solutions [41].

## E  Smoothness Assumption

Assumption 3.3 is a standard premise in graph signal processing and learning on graphs, closely related to the notion of homogeneity and homophily in datasets [42]. To assess how frequently this assumption holds in practice, we evaluate the $k$-smoothness of node features by computing the fraction of their energy captured by the subspace spanned by the first $k$ Laplacian eigenvectors. Table 5 reports the average energy concentration across all datasets used in our experiments for various values of $k$.

Notably, our method performs well even when the smoothness assumption is only partially satisfied. This highlights its robustness and broader applicability beyond the ideal smooth setting.

## F  Hyperparameters Discussion and Ablation Study

We review the purpose of each term in our optimization and clarify the motivations behind selecting the hyperparameters values. The parameter $\beta$ promotes minimizing the IPE for general smooth signals. As shown in Proposition 3, minimizing the respective term also bounds the REE (related to preserving the graph's global structure) and DE (related to preserving the norm of node features). Therefore, $\beta$ is significant when these properties are prioritized in coarsening. The parameter $\lambda$ enforces group sparsity in each row, ensuring the validity of the obtained coarsening operator $C$. Since $C$ lacks meaningful structure without this term, we did not perform an ablation study on $\lambda$. Finally, the parameter $\alpha$ promotes connectivity in the coarsened graph, making it significant in scenarios where preserving graph connectivity is essential.

Figure 4 presents an experiment analyzing each parameter's contribution and our method's sensitivity to their variations. The bars represent normalized scores for different values of a given hyperparameter, with distinct colors denoting specific values. For all metrics, lower values indicate better performance. The other two parameters are set to their optimal values for each metric as specified in Table 9. The figure illustrates the sensitivity of each parameter and evaluates the impact of deviations from optimal values on various metrics.

In Figures 4(a) and 4(b), varying $\alpha$ shows minimal sensitivity across metrics, except for IPE, where changes up to an order of magnitude still yield similar results. Additionally, the figures include an ablation study on the parameter $\alpha$ illustrating its contribution to the optimization process. In Figures 4(c) and 4(d), varying $\lambda$ demonstrates that our methods are more sensitive to this parameter, highlighting its critical role in performance.

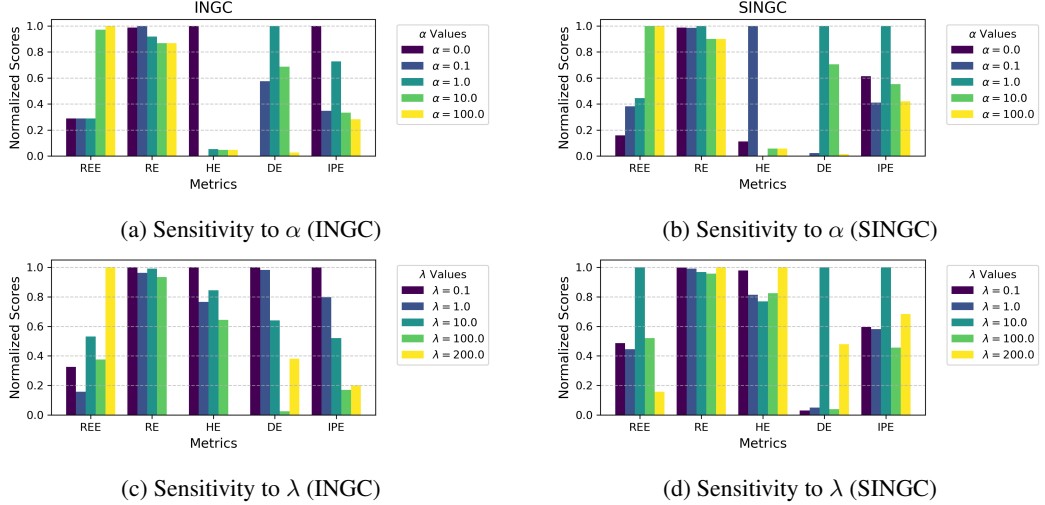

(a) Sensitivity to $\alpha$ (INGC)  (b) Sensitivity to $\alpha$ (SINGC)

(c) Sensitivity to $\lambda$ (INGC)  (d) Sensitivity to $\lambda$ (SINGC)

Figure 4: Ablation study on the sensitivity and contribution of the hyperparameters $\alpha$ and $\lambda$ across different metrics on the Cora dataset with a coarsening ratio $r = 0.3$. (a) and (b) show the sensitivity of the parameter $\alpha$ across metrics. (c) and (d) illustrate the sensitivity of the methods to parameter $\lambda$. The bars represent normalized scores for different values of the respective hyperparameter, with distinct colors denoting specific values. Lower bar values indicate better performance.

Table 6: Ablation study of the parameter $\beta$ on node classification tasks. The table reports the accuracy on various datasets for different coarsening ratios $r$ using different coarsening methods. The third column presents the results of our INGC method with $\beta = 0$, the fourth column corresponds to the optimal $\beta$ value, and the fifth column shows the results of SINGC. Best results are in bold; second-best results are underlined. The last two rows indicate the number of times each method achieved the best and second-best performance.

| Dataset | r | INGC($\beta = 0$) | INGC | SINGC |
|---|---|---|---|---|
| Cora | 0.3 | $84.62 \pm 0.59$ | $\mathbf{87.55 \pm 0.16}$ | $84.51 \pm 0.33$ |
| | 0.1 | $\underline{83.01 \pm 0.53}$ | $\mathbf{83.38 \pm 0.47}$ | $82.76 \pm 0.32$ |
| | 0.05 | $76.92 \pm 1.11$ | $\underline{77.42 \pm 0.78}$ | $\mathbf{77.81 \pm 0.68}$ |
| Citeseer | 0.3 | $76.25 \pm 0.28$ | $\mathbf{76.89 \pm 0.23}$ | $\underline{76.66 \pm 0.27}$ |
| | 0.1 | $67.07 \pm 0.59$ | $\mathbf{72.63 \pm 0.25}$ | $\underline{69.71 \pm 0.72}$ |
| | 0.05 | $60.66 \pm 1.58$ | $\underline{66.02 \pm 0.32}$ | $\mathbf{66.37 \pm 0.57}$ |
| Pubmed | 0.05 | $\mathbf{83.60 \pm 0.23}$ | $\underline{83.59 \pm 0.22}$ | $83.55 \pm 0.32$ |
| | 0.03 | $81.62 \pm 0.14$ | $\underline{81.93 \pm 0.22}$ | $\mathbf{83.19 \pm 0.18}$ |
| | 0.01 | $79.08 \pm 0.72$ | $\underline{79.09 \pm 0.26}$ | $\mathbf{79.96 \pm 0.34}$ |
| Co-CS | 0.05 | $90.42 \pm 0.18$ | $\underline{90.84 \pm 0.12}$ | $\mathbf{90.92 \pm 0.22}$ |
| | 0.03 | $89.28 \pm 0.21$ | $\underline{89.59 \pm 0.38}$ | $\mathbf{89.99 \pm 0.41}$ |
| | 0.01 | $77.79 \pm 1.15$ | $\mathbf{87.93 \pm 0.33}$ | $\underline{83.39 \pm 0.33}$ |
| #Best | | 1 | 6 | 6 |
| #2-Best | | 1 | 6 | 5 |

In Table 6, we present an ablation study on the parameter $\beta$ for the node classification task across various datasets and coarsening ratios $r$. The comparison includes three methods: INGC with $\beta = 0$ (ignoring the term $\mathrm{tr}(U^{(k)}(U^{(k)})^\top CC^\top)$ for minimizing IPE for general smooth signals), INGC with the optimal $\beta$, and SINGC (our second proposed method, which omits the first term of the objective entirely). The table reports node classification accuracy, with the best results highlighted in bold and the second-best results underlined. For each metric, other hyperparameters are set to their optimal values. The results demonstrate the importance of balancing the two complementary approaches to minimizing IPE. INGC with $\beta = 0$ generally underperforms compared to the other methods, emphasizing the significance of the smooth signal term in achieving high classification accuracy.

# G Additional Details on the Experimental Study

## Reproducibility Statement

We ensure reproducibility by providing a detailed description of our methodology, including algorithmic steps (Algorithms 1, 2), evaluation procedures, and hyperparameter settings (Appendix G.3,G.2). The code used in this paper will be made available in a public repository upon acceptance. Full proofs of the theoretical results are included in the appendix, along with precise descriptions of our experimental setups.

## G.1 Datasets Details

Table 7: Graph coarsening metrics experimental setting: Chosen hyperparameters for the Karate Club dataset at different coarsening ratios ($r$) and metrics.

| Metric | Method | Karate Club dataset | | |
|---|---|---|---|---|
| | | $r = 0.7$ | $r = 0.5$ | $r = 0.3$ |
| REE | INGC | $\beta=0, \lambda=100, \alpha=0.1$ | $\beta=200, \lambda=10, \alpha=1$ | $\beta=100, \lambda=100, \alpha=0.1$ |
| | SINGC | $\lambda=0.1, \alpha=0.1$ | $\lambda=0.1, \alpha=0.1$ | $\lambda=0.1, \alpha=0.1$ |
| RE | INGC | $\beta=0, \lambda=1, \alpha=200$ | $\beta=200, \lambda=0.1, \alpha=200$ | $\beta=200, \lambda=1, \alpha=200$ |
| | SINGC | $\lambda=0.1, \alpha=200$ | $\lambda=0.1, \alpha=200$ | $\lambda=1, \alpha=200$ |
| HE | INGC | $\beta=0, \lambda=1, \alpha=200$ | $\beta=200, \lambda=0.1, \alpha=200$ | $\beta=200, \lambda=1, \alpha=200$ |
| | SINGC | $\lambda=0.1, \alpha=200$ | $\lambda=0.1, \alpha=200$ | $\lambda=10, \alpha=200$ |
| DEE | INGC | $\beta=0, \lambda=1, \alpha=200$ | $\beta=200, \lambda=1, \alpha=200$ | $\beta=0.1, \lambda=1, \alpha=200$ |
| | SINGC | $\lambda=10, \alpha=200$ | $\lambda=1, \alpha=200$ | $\lambda=200, \alpha=200$ |

Table 8: Graph coarsening metrics experimental setting: Chosen hyperparameters for the Les Miserables dataset at different coarsening ratios ($r$) and metrics.

| Metric | Method | Les Miserables dataset | | |
|---|---|---|---|---|
| | | $r = 0.7$ | $r = 0.5$ | $r = 0.3$ |
| REE | INGC | $\beta=100, \lambda=200, \alpha=0.1$ | $\beta=200, \lambda=10, \alpha=0.1$ | $\beta=200, \lambda=200, \alpha=1$ |
| | SINGC | $\lambda=0.1, \alpha=0.1$ | $\lambda=100, \alpha=0.1$ | $\lambda=0.1, \alpha=0.1$ |
| RE | INGC | $\beta=200, \lambda=10, \alpha=100$ | $\beta=200, \lambda=10, \alpha=200$ | $\beta=200, \lambda=100, \alpha=200$ |
| | SINGC | $\lambda=0.1, \alpha=100$ | $\lambda=1, \alpha=200$ | $\lambda=100, \alpha=100$ |
| HE | INGC | $\beta=200, \lambda=10, \alpha=100$ | $\beta=200, \lambda=10, \alpha=200$ | $\beta=200, \lambda=100, \alpha=200$ |
| | SINGC | $\lambda=0.1, \alpha=100$ | $\lambda=0.1, \alpha=200$ | $\lambda=100, \alpha=100$ |
| DEE | INGC | $\beta=0, \lambda=200, \alpha=200$ | $\beta=0.1, \lambda=0.1, \alpha=10$ | $\beta=0.1, \lambda=0.1, \alpha=200$ |
| | SINGC | $\lambda=100, \alpha=100$ | $\lambda=10, \alpha=100$ | $\lambda=100, \alpha=200$ |

## G.2 Node Classification Experiments Setting

The GCN model used in our experiments consists of two graph convolutional layers and is implemented using PyTorch and PyTorch Geometric libraries. The architecture is as follows:

- **Layer 1:** A Graph Convolutional Network (GCNConv) layer that takes the input node feature matrix $X$ (with $X.shape[1]$ features) and outputs a hidden representation of size 64.
- **Layer 2:** A second GCNConv layer that maps the 64-dimensional hidden representation to the number of output classes (`NUM_OF_CLASSES`).

We use ReLU for non-linearity and dropout for regularization during training.

Table 9: Graph coarsening metrics experimental setting: Chosen hyperparameters for the Cora dataset at different coarsening ratios ($r$) and metrics.

| Metric | Method | Cora dataset | | |
| --- | --- | --- | --- | --- |
| | | $r = 0.7$ | $r = 0.5$ | $r = 0.3$ |
| REE | INGC | $\beta$=10, $\lambda$=1, $\alpha$=1 | $\beta$=100, $\lambda$=1, $\alpha$=1 | $\beta$=200, $\lambda$=10, $\alpha$=10 |
| | SINGC | $\lambda$=1, $\alpha$=10 | $\lambda$=100, $\alpha$=100 | $\lambda$=200, $\alpha$=0.1 |
| RE | INGC | $\beta$=10, $\lambda$=100, $\alpha$=200 | $\beta$=100, $\lambda$=200, $\alpha$=200 | $\beta$=100, $\lambda$=10, $\alpha$=200 |
| | SINGC | $\lambda$=100, $\alpha$=200 | $\lambda$=0.1, $\alpha$=100 | $\lambda$=100, $\alpha$=100 |
| HE | INGC | $\beta$=10, $\lambda$=100, $\alpha$=200 | $\beta$=100, $\lambda$=200, $\alpha$=200 | $\beta$=100, $\lambda$=10, $\alpha$=200 |
| | SINGC | $\lambda$=100, $\alpha$=200 | $\lambda$=1, $\alpha$=10 | $\lambda$=100, $\alpha$=100 |
| DEE | INGC | $\beta$=0.1, $\lambda$=0.1, $\alpha$=10 | $\beta$=0.1, $\lambda$=100, $\alpha$=200 | $\beta$=0, $\lambda$=10, $\alpha$=10 |
| | SINGC | $\lambda$=100, $\alpha$=200 | $\lambda$=10, $\alpha$=100 | $\lambda$=100, $\alpha$=200 |

Table 10: Graph coarsening metrics experimental setting: Chosen hyperparameters for the Citeseer dataset at different coarsening ratios ($r$) and metrics.

| Metric | Method | Citeseer dataset | | |
| --- | --- | --- | --- | --- |
| | | $r = 0.7$ | $r = 0.5$ | $r = 0.3$ |
| REE | INGC | $\beta$=200, $\lambda$=200, $\alpha$=200 | $\beta$=100, $\lambda$=10, $\alpha$=0.1 | $\beta$=100, $\lambda$=10, $\alpha$=0.1 |
| | SINGC | $\lambda$=10, $\alpha$=0.1 | $\lambda$=100, $\alpha$=0.1 | $\lambda$=10, $\alpha$=1 |
| RE | INGC | $\beta$=100, $\lambda$=10, $\alpha$=200 | $\beta$=100, $\lambda$=1, $\alpha$=10 | $\beta$=0, $\lambda$=10, $\alpha$=0.1 |
| | SINGC | $\lambda$=1, $\alpha$=100 | $\lambda$=10, $\alpha$=100 | $\lambda$=100, $\alpha$=200 |
| HE | INGC | $\beta$=100, $\lambda$=10, $\alpha$=200 | $\beta$=0.1, $\lambda$=100, $\alpha$=200 | $\beta$=200, $\lambda$=0.1, $\alpha$=0.1 |
| | SINGC | $\lambda$=1, $\alpha$=100 | $\lambda$=1, $\alpha$=100 | $\lambda$=100, $\alpha$=200 |
| DEE | INGC | $\beta$=1, $\lambda$=0.1, $\alpha$=0.1 | $\beta$=200, $\lambda$=1, $\alpha$=10 | $\beta$=0.1, $\lambda$=0.1, $\alpha$=10 |
| | SINGC | $\lambda$=100, $\alpha$=200 | $\lambda$=100, $\alpha$=200 | $\lambda$=100, $\alpha$=100 |

For SINGC, we set $t_{\text{iter}} = 2000$ in all experiments, and for INGC, we set $t_{\text{iter}} = 20$ and $c_{\text{iter}} = 100$. Tables 12 and 11 present the hyperparameters of our methods for each experiment. The results for the three baseline methods presented in Table 2 are sourced from Kumar et al. [14].

We note that tuning the hyperparameters in our methods is crucial for achieving optimal performance. By reviewing some of the corresponding setting in Tables 10, 9 and 11 we can observe that good performance often aligns with low values of REE and INP in this application. Therefore, we recommend that practitioners first optimize the hyperparameters by minimizing REE and INP. Once optimized, the coarsened graph can be used in the GNN for training and evaluation, leading to improved classification accuracy.

The additional details of real datasets are as follows:

Table 11: Node classification experimental setting: Chosen hyperparameters for the Cora Citeseer dataset at different coarsening ratios ($r$) and metrics.

| Dataset | Method | Node Classification Parameters - Medium datasets | | |
| --- | --- | --- | --- | --- |
| | | $r = 0.3$ | $r = 0.1$ | $r = 0.05$ |
| Cora | INGC | $\beta$=100, $\lambda$=100, $\alpha$=10 | $\beta$=1, $\lambda$=100, $\alpha$=1 | $\beta$=1, $\lambda$=100, $\alpha$=100 |
| | SINGC | $\lambda$=10, $\alpha$=0.01 | $\lambda$=1000, $\alpha$=1 | $\lambda$=1000, $\alpha$=0.01 |
| Citeseer | INGC | $\beta$=0, $\lambda$=100, $\alpha$=10 | $\beta$=10, $\lambda$=1000, $\alpha$=1000 | $\beta$=0, $\lambda$=20, $\alpha$=10 |
| | SINGC | $\lambda$=50, $\alpha$=20 | $\lambda$=300, $\alpha$=100 | $\lambda$=50, $\alpha$=10 |

Table 12: Node classification experimental setting: Chosen hyperparameters for the Co-phy, Pubmed, Co-CS dataset at different coarsening ratios ($r$) and metrics.

| Dataset | Method | Node Classification Parameters - Large datasets | | |
| --- | --- | --- | --- | --- |
| | | $r = 0.05$ | $r = 0.03$ | $r = 0.01$ |
| Co-phy | INGC | $\beta$=10, $\lambda$=10, $\alpha$=0.01 | $\beta$=10, $\lambda$=1000, $\alpha$=0.01 | $\beta$=10, $\lambda$=1000, $\alpha$=100 |
| | SINGC | $\lambda$=100, $\alpha$=0.01 | $\lambda$=10, $\alpha$=1 | $\lambda$=10, $\alpha$=100 |
| Pubmed | INGC | $\beta$=0, $\lambda$=1000, $\alpha$=0.001 | $\beta$=0.1, $\lambda$=100, $\alpha$=10 | $\beta$=0.1, $\lambda$=100, $\alpha$=100 |
| | SINGC | $\lambda$=1000, $\alpha$=0.001 | $\lambda$=100, $\alpha$=0.1 | $\lambda$=10, $\alpha$=10 |
| Co-CS | INGC | $\beta$=1, $\lambda$=1000, $\alpha$=10 | $\beta$=0.1, $\lambda$=1, $\alpha$=10 | $\beta$=10, $\lambda$=100, $\alpha$=100 |
| | SINGC | $\lambda$=40, $\alpha$=10 | $\lambda$=10, $\alpha$=10 | $\lambda$=100, $\alpha$=100 |

- **Karate Club** - $n = 34$, $p = 30$, $|\mathcal{E}| = 78$ - Here, nodes represent members of a karate club, and edges represent friendships between them. Synthetic features generated using the signal model presented at Section 3.1.

- **Les Miserables** - $n = 77$, $p = 50$, $|\mathcal{E}| = 254$ - Nodes represent characters in the novel *Les Miserables*, and edges indicate co-occurrence in the same chapter. Synthetic features generated using the signal model presented at Section 3.1.

- **Cora** - $n = 2,708$, $p = 1,433$, $|\mathcal{E}| = 5,429$ - Nodes represent research papers, and edges represent citation links between them. Node features correspond to the presence of specific words in each paper, and class labels indicate the paper's research field. Number of classes = 7.

- **Citeseer** - $n = 3,327$, $p = 3,703$, $|\mathcal{E}| = 4,732$ - Nodes represent research papers, and edges represent citation relationships. Node features are based on word occurrences in each paper, and class labels indicate the paper's topic. Number of classes = 6.

- **Co-Physics** - $n = 34,493$, $p = 8,415$, $|\mathcal{E}| = 247,962$ - Nodes represent physics research papers, and edges represent citations. Node features represent article keywords, and class labels indicate different fields of physics. Number of classes = 5.

- **PubMed** - $n = 19,717$, $p = 500$, $|\mathcal{E}| = 44,338$ - Nodes represent biomedical research papers, and edges represent citations. Node features are derived from TF-IDF scores of medical terms, and class labels indicate disease categories. Number of classes = 3.

- **Co-Computer** - $n = 13,752$, $p = 767$, $|\mathcal{E}| = 245,861$ - Nodes represent products in a co-purchase network, and edges indicate products frequently purchased together. Node features describe product attributes, and class labels represent product categories. Number of classes = 10.

- **Co-CS** - $n = 18,333$, $p = 7005$ , $|\mathcal{E}| = 163,788$ - Here, nodes are authors, that are connected by an edge if they co-authored a paper; node features represent paper keywords for each author's papers, and class labels indicate most active fields of study for each author. Number of classes = 15.

## G.3 Graph Coarsening Metrics Experiments Setting

Tables 7, 8, 9, and 10 present the hyperparameters of our methods for each experiment. For SINGC, we set $t_{\text{iter}} = 2000$ in all experiments, and for INGC, we set $t_{\text{iter}} = 20$ and $c_{\text{iter}} = 100$.

Regarding the implementation of the baseline comparison methods, the FGC hyperparameters were selected based on their optimal values as reported in their paper. The LVN and LVE methods were implemented using their provided graph coarsening libraries, with the maximum value of the parameter $K = k = r \cdot n$.

**Karate Club**

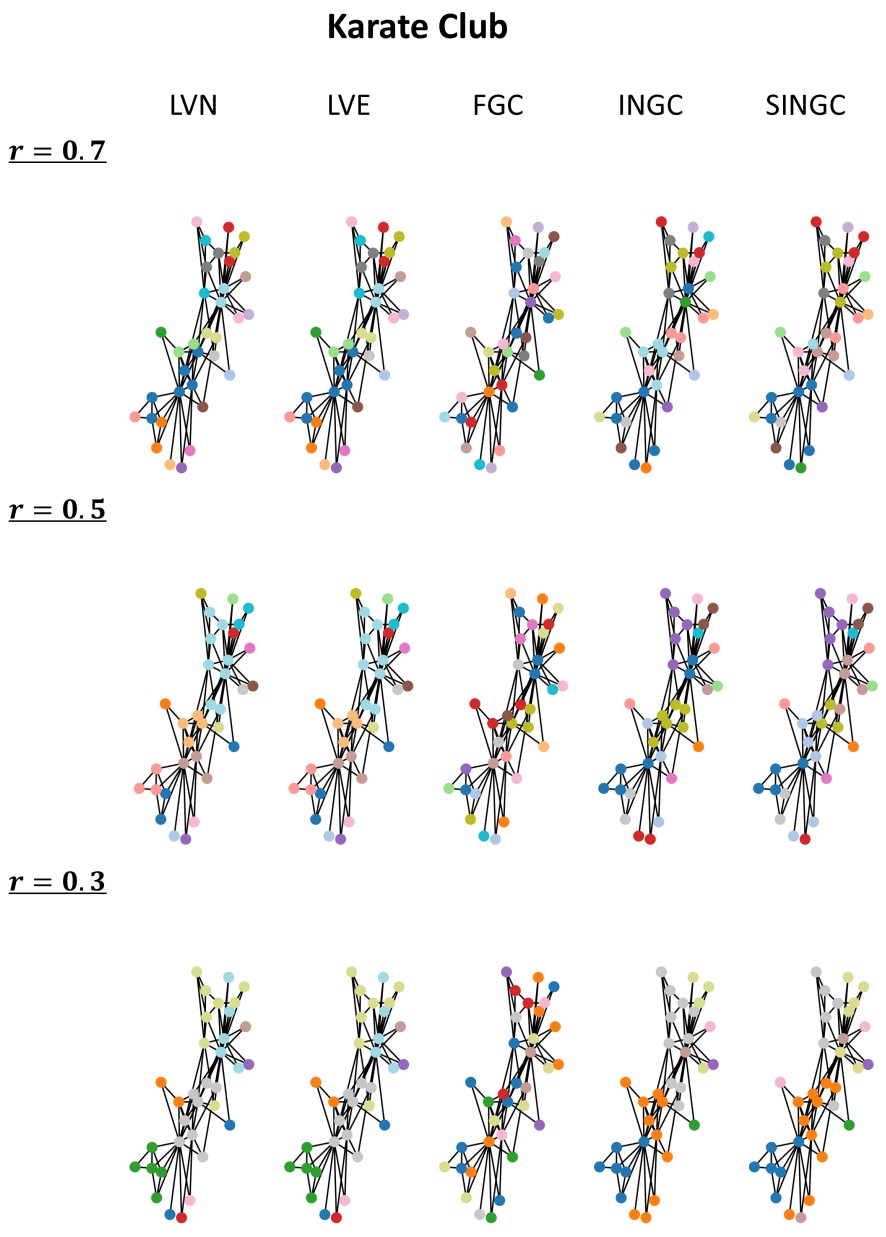

Figure 5: Visual comparison of coarsening methods on the Karate Club dataset. Each row displays the partitioning produced by each method at a different coarsening ratio. Nodes of the same color are grouped into the same super-node.

# Les Miserables

LVN      LVE      FGC      INGC      SINGC

$r = 0.7$

$r = 0.5$

$r = 0.3$

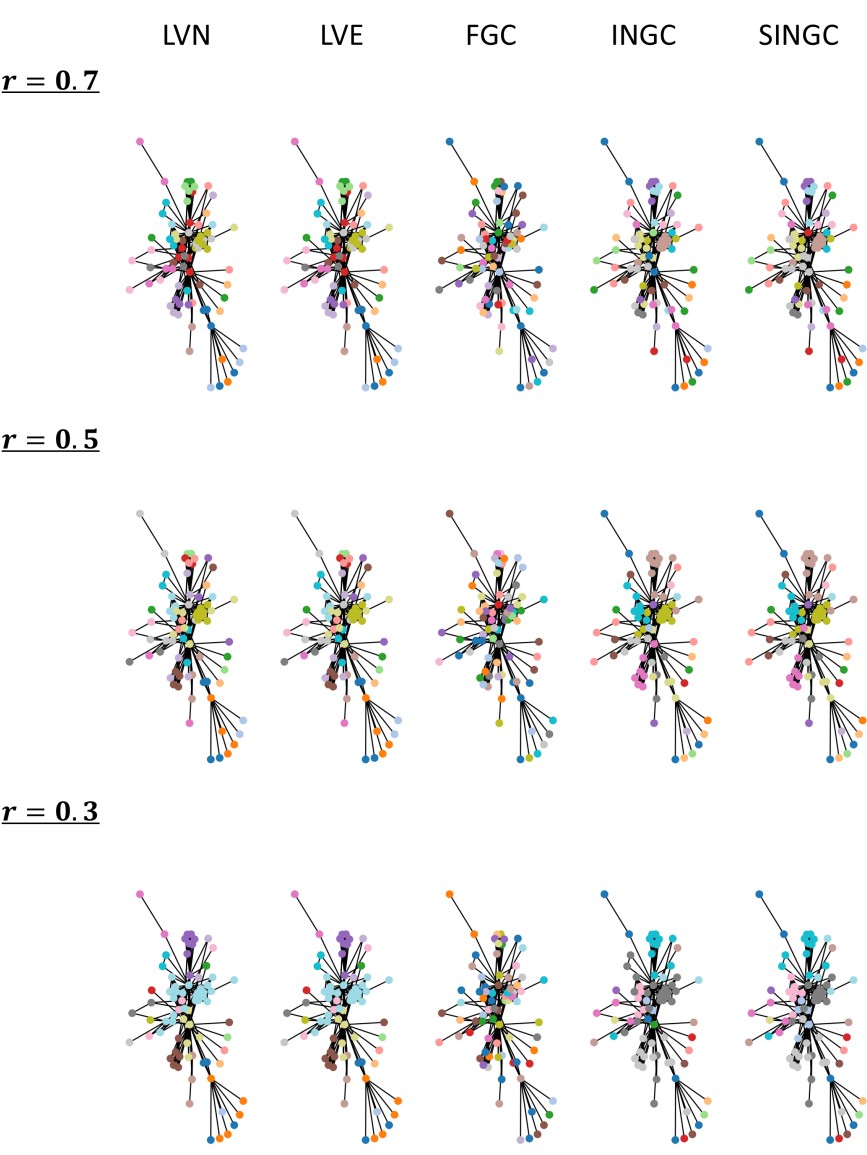

Figure 6: Visual comparison of coarsening methods on the Les Miserables dataset. Each row displays the partitioning produced by each method at a different coarsening ratio. Nodes of the same color are grouped into the same super-node.

# Planner Graph

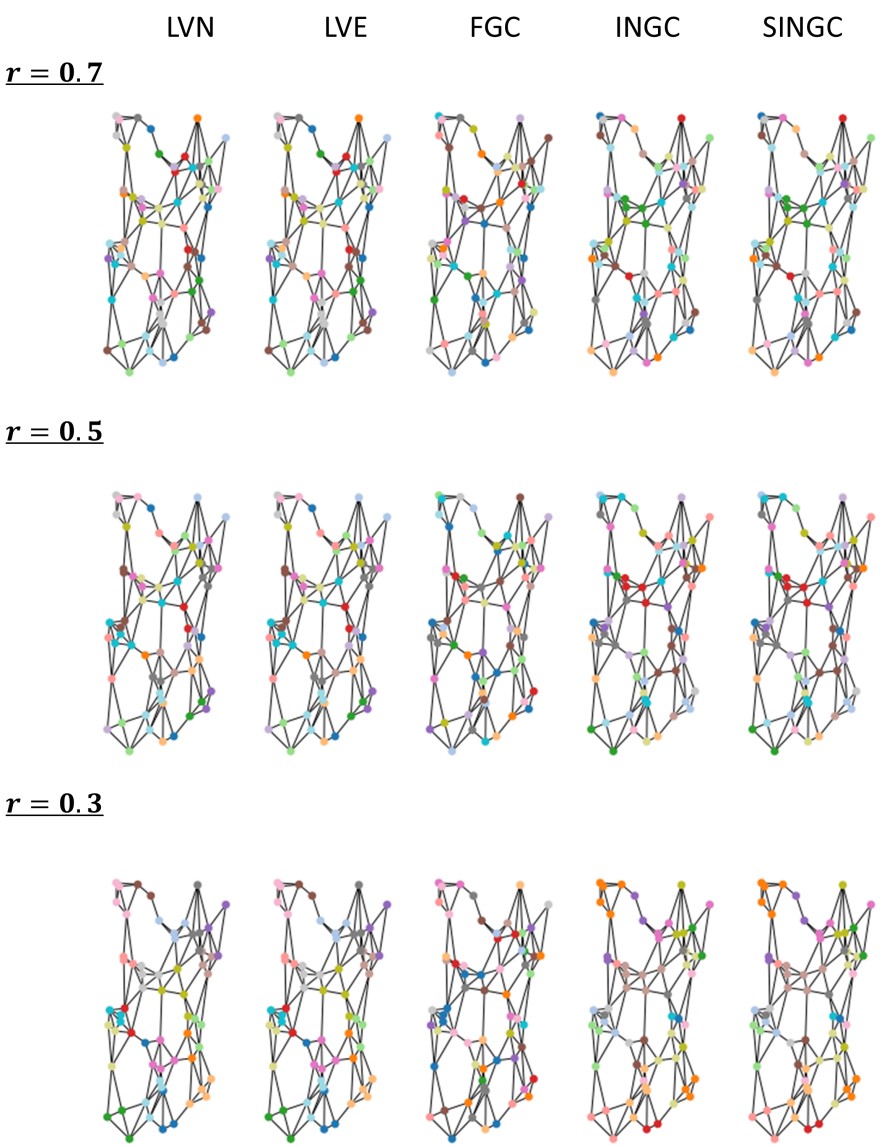

Figure 7: Visual comparison of coarsening methods on a planner graph. Each row displays the partitioning produced by each method at a different coarsening ratio. Nodes of the same color are grouped into the same super-node.

