# OpenReview forum: "Spectral Graph Coarsening Using Inner Product Preservation and the Grassmann Manifold"
_NeurIPS.cc/2025/Conference — NeurIPS 2025 poster_

### Official Review · Reviewer_sNPx · 2025-06-22

**Clarity:** 2
**Significance:** 2
**Originality:** 2
**Rating:** 4
**Confidence:** 3

**Summary:**

This paper introduces a new graph coarsening method for node-featured graphs. For this, the authors define the Inner-Product Error (IPE) $||X^\top LX - X_c^\top L_c X_c ||_F^2$ as a metric that captures how well coarsened graphs preserve the Laplacian-induced inner products of node features. The authors design a minimization objective that couples IPE with a Grassmann-alignment term encouraging coarsening maps to be aligned with the top $k$ Laplacian eigenvectors of the graph, and they propose two coarsening algorithms, the feature-aware INGC and the feature-agnostic SINGC. They then provide some theoretical justification for their choice of objective. Finally, they demonstrate superiority of their method through an empirical study, comparing it to common coarsening methods (local variation neighborhood/edges, feature-based graph coarsening), as well as downstream performance on standard node classification tasks.

**Questions:**

- Regarding the Laplacian regularization term in the objective in Eq. (9), in lines 179-180 you state that this "guarantees that the coarsened graph $\mathcal{G}_c$ is connected". Why enforce connectivity unconditionally? Does your method assume the original graph is connected, and if not, wouldn’t this merge naturally disconnected components and reduce generality? I would appreciate if you could clarify the rationale.
- In the beginning of § 3.1/Eq. (10), a "smooth graph signal generation mechanism" is introduced, but it seems that this is never really mentioned again later, as Assumption 3.3 proceeds with defining such signals simply as being exactly in the column space of the top $k$ eigenvectors. Is there any point in stating it, other than motivating Assumption 3.3?
- In several of the visualizations (like pages 29/30) it is quite difficult to visually discern which coarsening method performs better. Perhaps including more structured or (close to) planar graphs could help. Also, since one of the method's key selling points is preserving node-feature relationships, a small example that actually displays how features are mapped could greatly improve intuition.
- A few times you mention your method is "functorial". Is this a standard term in this context that I'm not aware of? When skimming the abstract for the first time I thought there might be some connection to category theory, but it seems you are using the term more loosely to mean "functional" or "structure-preserving". I might suggest tweaking the wording.

**Ethical Concerns:**

["NO or VERY MINOR ethics concerns only"]

**Final Justification:**

The authors addressed most of my concerns, and with more visualizations and improved presentation I think this paper could be accepted. Yet, in my view, the feature smoothness assumption remains a notable limitation of the work as is, which is why I did not change my score.

**Limitations:**

yes

**Paper Formatting Concerns:**

*Minor* formatting issues:
- proofs in the appendix are written in a theorem environment, maybe use `\begin{proof}...\end{proof}` instead.
- Propositions 3.4 and 3.5 are referred to as "theorems" in the appendix.
- lines 219/220: superscript should be `(i)`.

**Quality:**

3

**Strengths And Weaknesses:**

**Strengths:**
- Incorporating node features into graph coarsening methods is important as it is widely acknowledged that node features can be just as important for downstream tasks as the actual graph structure. To my knowledge, the introduced IPE metric is novel.
- The empirical results (e.g, Table 1) look compelling, as INGC and SINGC (i.e., even not taking into account node features and using just the Grassmannian term) improve standard coarsening methods more or less across the board.

**Weaknesses:**
- I feel like the theoretical justification is somewhat limited, as, e.g., Proposition 3.4 assumes *perfectly* $k$-smooth features and a coarsening operator $C$ that *precisely* spans the column space of the top $k$ eigenvectors. Also, technical novelty is rather modest, e.g., Proposition 3.5 is a reduction to Thm 13 of [1].
- The design and theoretical justifications for the method hinge on spectral smoothness of the node features, which real-world graphs might violate. The authors briefly discuss this in § E of the appendix. While they show that standard datasets like Cora, Citeseer etc. fulfill this approximately, I would assume that this would struggle with more heterophilic graphs where node features would also typically be of high frequency. It could strengthen the work to analyze more thoroughly when/how feature non-smoothness actually degrades the coarsening quality.
- The presentation and clarity could be improved, as not all details were immediately clear to me. For example, it was not obvious to me from the text description what constraint one would project onto with PSGD in INGC. I am adding a few other (though minor) suggestions to improve the clarity under "Questions".

[1] Andreas Loukas. *Graph Reduction with Spectral and Cut Guarantees.* JMLR 20 (2019) 1-42.

---

> ### Author Rebuttal · Authors · 2025-07-30
>
> Thank you for the time and effort you put into reviewing our paper. Your
> comments were very constructive and helped us improve the manuscript. Our responses to the main concerns you raised are:
>
> # *Theoretical Contribution*
> Our theoretical justification is composed of three components that together support the design of our method.
>
> (i) Proposition 3.2 provides a conceptual foundation for minimizing the IPE as a meaningful proxy for preserving graph structure during coarsening. While we acknowledge that the conditions under which this proposition holds—such as the rank constraints—are unrealistic in most practical settings, the result is not intended to suggest that exact Laplacian reconstruction is feasible. Rather, it serves to motivate the use of IPE minimization as a principled structural preservation objective.
>
> (ii) Proposition 3.4 shows that maximizing the Grassmann similarity between the coarsening operator and the subspace spanned by the top-$k$ Laplacian eigenvectors leads to IPE minimization for graph signals that satisfy the smoothness assumption. This observation directly motivates the second term in our objective function, which promotes subspace alignment on the Grassmann manifold. As a result, our method ensures low IPE not only for observed features but also for unseen smooth signals.
>
> (iii) Proposition 3.5 connects this subspace alignment to improvements in widely used graph coarsening metrics such as REE and DEE. Although the proof of Proposition 3.5 leverages Theorem 13 from [1], its combination with Proposition 3.4 provides a stronger theoretical insight: that minimizing IPE for general smooth signals leads to improved structural preservation. We believe this connection offers a principled geometric justification for the effectiveness of our approach.
>
> # *Smoothness Assumption*
>
> A central assumption in our work is that node features are smooth with respect to the graph structure, which is connected to the graph homophily assumption. This smoothness assumption underlies both our formulation and theoretical guarantees. For this reason, we do not include heterophilic datasets in our experimental evaluation.
>
> More specifically, the second term in our objective relies on this smoothness assumption, as it encourages alignment with the subspace spanned by the low-frequency eigenvectors of the graph Laplacian. In heterophilic graphs, this assumption may not hold, and our theoretical results would no longer apply.
>
> One potential direction for adapting our method to such settings is to modify the second term in the objective to project onto a different frequency range—e.g., using $\boldsymbol{U}^{(k)}$ corresponding to mid- or high-frequency eigenvectors if prior knowledge about label-frequency alignment is available. However, such a modification would require a fundamentally different theoretical treatment, which we believe is beyond the scope of this work.
>
> Following your suggestion, we have emphasized this limitation and possible extensions in the Conclusion section of the revised manuscript.
>
> We note that our empirical results indicate that our proposed methods remain competitive even when the smoothness assumption is only weakly satisfied. For example, Table 5 in Appendix E shows that when roughly 25\% of a signal’s energy lies in the low-frequency spectrum—as in the Cora, Citeseer, and Co-CS datasets at $r = 0.05$—our method still matches state-of-the-art baselines.
>
> Moreover, Table 5 and our main experimental results (Table 2) illustrate that the advantage of SINGC becomes more pronounced as a larger fraction of signal energy resides in the low-frequency band. For instance, on the Pubmed dataset, where about 35\% of the energy is concentrated in the relevant spectrum, SINGC surpasses all competing baselines, highlighting its advantage in such scenarios. Following the reviewer's comment, we clarified this property at the end of Section 4.
>
> # *Demonstrative Visualizations*
>
> We thank the reviewer for the helpful suggestions regarding the visualizations.
>
> Following your comment, we added an additional visualization using a planar graph with 40 vertices, where all methods are applied. The clustering behavior of our method is indeed more visually clear in this example(unfortunately, we cannot share it here due to format constraints). The figures demonstrate the "cluster-preserving" properties of our approach—despite being guided by feature similarity rather than explicit structural information, our method produces clusters that closely resemble those generated by structure-based methods such as LVE and LVN, which do not use node features. This suggests that our method is capable of capturing meaningful structural patterns through the lens of feature similarity.
> We also note that this property is further illustrated in Figure 2 in the Appendix B.
>
> In addition, we will incorporate a new figure in Appendix B to visually demonstrate the preservation of node-feature relationships. Specifically, we compare two heatmaps:
> \begin{align*}
> H = \boldsymbol{X}^\top \boldsymbol{L} \boldsymbol{X} \quad and \quad H_c = \boldsymbol{X}_c^\top \boldsymbol{L}_c \boldsymbol{X}_c,
> \end{align*}
>
> where the $(s,k)$-th entry of $H$ (and similarly for $H_c$) is given by:
>
> \begin{align*}
> H[s,k]=\boldsymbol{x}^{(s)\top} \boldsymbol{L} \boldsymbol{x}^{(k)} = \sum_{(i,j) \in E} w_{ij} \left( x^{(s)}(i) - x^{(s)}(j) \right)\left( x^{(k)}(i) - x^{(k)}(j) \right),
> \end{align*}
>
> and $\boldsymbol{x}^{(s)} = \boldsymbol{X}[s,:]$ and $\boldsymbol{x}^{(k)} = \boldsymbol{X}[k,:]$ are the $s$-th and $k$-th node feature vectors, respectively.
>
> This comparison—performed on our SBM experiment — visually demonstrates how our method preserves the original feature relationships. We note that, by the definition of the coarsening matrix $\boldsymbol{P}$, the super-node features are formed as the average of their respective groups, further supporting the intuitive interpretation of feature aggregation under our framework.
>
> # *Additional clarification*
>  1.) In INGC (Algorithm 1), the projection step is performed every $c_{\text{iter}}$ iterations—specifically, before each update of $\boldsymbol{X}_c$. This is done using the Hardmax$(\boldsymbol{C})$ operator, which projects $\boldsymbol{C}$ onto the constraint set $\mathcal{C}$ by enforcing a column-wise orthogonality constraint consistent with the Grassmann manifold structure. We agree that this design could be made clearer in the current algorithm description, and we will revise Algorithm~1 in the updated manuscript by inserting the projection step between Lines 8 and 9 and explicitly referencing the Hardmax operator.
>
>  2.) We chose to adopt the same regularization terms as FGC to ensure a fair and controlled comparison, allowing us to isolate and evaluate the contribution of our proposed IPE term and overall coarsening approach. This includes the connectivity regularization term, which is controlled by the hyperparameter $\alpha$. Importantly, $\alpha$ can be adjusted based on the desired application.
>
> 3.)  Equation (10) is presented for two main reasons. The first, as you correctly noted, is to motivate Assumption 3.3 by illustrating that assuming signals lie in the span of the top $k$ Laplacian eigenvectors is standard in graph signal processing.
> Second, we use this signal generation model to construct node features in datasets where such features are not available- specifically in the toy example, the Karate Club graph, and the Les Misérables network.
>
> 4.) We use the term “functorial” to explain that our method operates from a function-based perspective. Following your suggestion, we will revise the wording to use “functional” instead, in order to avoid confusion and improve clarity.
>
> 5.) Thank you for pointing out our formatting issues. We accept all your suggestions and will fix them in the revised version of the paper.

---

> > ### Comment · Reviewer_sNPx · 2025-08-04
> > **Response by Reviewer**
> >
> > I thank the authors for their thoughtful rebuttal which has addressed the majority of my concerns. I appreciate the clarifications regarding the theoretical framing. I agree that, while idealized, the assumptions are clearly stated and the authors have acknowledged their limitations appropriately. I remain convinced that adding more demonstrative visualizations would significantly improve the accessibility and impact of the work. I am confident that the visualizations that the authors proposed will help and I hope they will be included in the camera-ready version. Regarding the spectral smoothness assumption: I still view this as a central limitation of the method’s applicability; yet, the authors' decision to scope their analysis to a homophilic setting is reasonable. I will keep my score unchanged.

---

### Official Review · Reviewer_9tpS · 2025-06-29

**Clarity:** 3
**Significance:** 3
**Originality:** 4
**Rating:** 5
**Confidence:** 4

**Summary:**

The paper proposes a novel graph coarsening method based on a similarity metric in the Grassmann manifold and the inner product defined in graph theory. The authors analyze the principle of the proposed method solidly, and compare it with the previous method in theory. It provides a geometric view to consider the graph coarsening problem. The extensive experiments show the effectiveness of the proposed INGC and SINGC.

**Questions:**

1. Can you make a further discussion about that if a graph truly has $n-k$ connected components, will the proposed method achieve the zero error theoretically?
2. The weights in coarsening matrix $C$ are soft or hard, i.e., the element ranges in $[0, 1]$ or $\\{0, 1 \\}$ ?
3. Can you discuss how the Grassmann manifold plays a vital role in the proposed methods?
4. Others refer to the **Weakness**.

**Ethical Concerns:**

["NO or VERY MINOR ethics concerns only"]

**Final Justification:**

After discussing with authors, my concerns are already addressed. Especially for the gradient projection process in learning the assignment matrix, the authors provide detailed clarification about algorithms. In all, I think the originality and novelty of this work are great. It proposes a new framework for graph coarsening that leverages the properties of the Grassmann manifold, which will promote the development of Riemannian graph learning to a certain extent.

**Limitations:**

Yes.

**Paper Formatting Concerns:**

No, there are no major formatting issues.

**Quality:**

3

**Strengths And Weaknesses:**

Pros:
1. The view that considers the coarsening problem with the Grassmann manifold has novelty and insight. It proposed a new coarsening metric.
2. The authors make some reasonable assumptions, making theoretical analysis more solid.
3. The experiments are detailed and convincing, since the authors analyze the model in detail by ablation study and visualization, even validating the sense of assumption.

Cons:
1. In Assumption 3.3 of the $k$-smooth graph signal, I think there is a typo that $x=c^T U^{(k)}$, the correct formula is $x=U^{(k)}c$, since the vector $c$ may be a column vector. Otherwise the dimension is mismatched, and $x=U^{(k)}(U^{(k)})^Tx$ will not hold in Appendix A.2, line 779.
2. Although the proposed method is solid in theory, I am concerned about the assumption for the $(n-k)$-connected components. Since for a large-scale graph, the inter-community structure is also important, but the proposed method is based on the situation for $(n-k)$-connected components, which may lose information about inter-community connecting patterns.
3. I think the usage of the Grassmann manifold in the gradient descent (GD) algorithm has a misunderstanding. I do not find any projection on gradients. Since in Eq. 8, you think the coarsening matrix $C$ is also in the Grassmann manifold, but as you apply GD to the point in the Grassmann manifold, if you do not project gradients, you can not guarantee that the gradient vector is in the tangent space, making the updated $C$ destroy the orthogonal property.

---

> ### Author Rebuttal · Authors · 2025-07-30
>
> Thank you for the time and effort you put into reviewing our paper. Your
> comments were very constructive and helped us improve the manuscript. Our responses to the main concerns you raised are:
>
> # *Theoretical Results Clarifications*
> The main purpose of Proposition 3.2 is to provide theoretical motivation for focusing on minimizing the IPE as a meaningful proxy for preserving the graph’s structure during coarsening. As noted, exact recovery of the original Laplacian $\boldsymbol{L} \in \mathbb{R}^{n \times n}$ from a coarsened Laplacian $\boldsymbol{L}_c \in \mathbb{R}^{k \times k}$ with $k \ll n$ is only possible under highly specific conditions, such as when the graph has more than $n - k$ connected components. However, the proposition highlights that preserving pairwise signal inner products during coarsening can help retain important structural properties of the original graph.
>
> Importantly, as also noted by another reviewer, we do not rely on this result in our formal proof. Thus, the condition in Proposition 3.2 can be relaxed to the following statement:
> "If the inner product is preserved for all pairs of graph signals, i.e., $\boldsymbol{x}^\top \boldsymbol{L} \boldsymbol{y} = \boldsymbol{x}_c^\top \boldsymbol{L}_c \boldsymbol{y}_c$, then the original Laplacian $\boldsymbol{L}$ can be exactly recovered from $\boldsymbol{L}_c$."
> However, a necessary condition for the first to hold is that the rank of $\boldsymbol{L}$ is at most $k$ (implying $n-k$ connected components).
>
> Following your comment, we will revise the manuscript to reflect this relaxed version of Proposition 3.2 and include a remark noting the required rank condition. We will also clarify that, although this assumption is unrealistic in most practical settings, the result is not intended to suggest that exact reconstruction is feasible in general. Rather, it serves to conceptually motivate the use of IPE minimization as a principled structural preservation objective in graph coarsening.
>
> # *The Role of the Grassmann Manifold*
> Both the coarsening matrix $\boldsymbol{C}$ and the matrix of leading Laplacian eigenvectors $\boldsymbol{U}^{(k)}$ can be viewed as points on the Grassmann manifold, as they represent $k$-dimensional subspaces in $\mathbb{R}^n$. In Proposition 3.4, we establish that maximizing the Grassmann similarity between these subspaces leads to minimization of the IPE for any pair of graph signals that satisfy the smoothness assumption. This insight directly motivates the second term in our objective, which explicitly promotes subspace alignment on the Grassmann manifold—thereby ensuring low IPE not only for observed features but also for unseen smooth signals.
>
> Furthermore, Propositions 3.4 and 3.5 together suggest that this subspace alignment provides theoretical guarantees for standard graph coarsening metrics. Specifically, minimizing IPE for general signals under the smoothness assumption leads to improvements in REE and DEE, offering a principled geometric justification for our method’s effectiveness.
>
> # *Optimization process*
> We thank the reviewer for raising this important point regarding the usage of the Grassmann manifold and projection steps in our gradient-based optimization.
>
> You are correct that, in general, gradient descent updates may move the coarsening matrix $\boldsymbol{C}$ outside the Grassmann manifold, and projection is needed to maintain the orthogonality constraint. In our approach, we follow a projected gradient descent scheme, where we periodically project the matrix $\boldsymbol{C}$ back onto the manifold using the operator Hardmax$(\boldsymbol{C})$. This operator maps $\boldsymbol{C}$ back to the Grassmann manifold by enforcing a column-wise orthogonality constraint consistent with the Grassmann manifold structure.
>
> In INGC Algorithm 1, this projection/mapping step is performed after every $c_{\text{iter}}$ iterations—specifically, before each update of $\boldsymbol{X}_c$. We agree that this design could be made clearer in the current algorithm description, and we will revise Algorithm 1 in the updated manuscript by inserting the projection step between Lines 8 and 9, and explicitly referring to the "Hardmax" operator.
>
> For SINGC Algorithm 2, we empirically observed that projecting only once at the end of the optimization yields performance similar to using intermediate projections. Therefore, for simplicity and efficiency, we chose to use standard gradient descent and apply the projection step only at the end in this case. We will add this clarification in the updated manuscript.
>
> We thank the reviewer again for pointing this out and helping us improve the clarity and correctness of the manuscript.
>
> # *Additional clarification*
> 1.) You are correct, there is a typo in Assumption 3.3, the correct formula that aligns with our notation is $x=c (U^{(k)})^\top$.
>
> 2.) The matrix $\boldsymbol{C}$ applies "hard coarsening", with elements in {0, 1}. We denote this assignment using the Hardmax$(\boldsymbol{C})$ operator, which is applied in the last line of both of our proposed algorithms.

---

> > ### Comment · Reviewer_9tpS · 2025-08-03
> > **Official comment by Reviewer 9tpS**
> >
> > Dear authors, thanks for your active response. Most concerns are addressed. But importantly, the projection in gradient descent—how will you add this clarification for Algorithms 1 and 2 in the updated manuscript? Please give me some detailed description; it is better to add some mathematical formulas. I will raise the rating if this problem can be clearly solved.

---

> ### Author Response · Authors · 2025-08-03
> **Followup response**
>
> We thank the reviewer for their follow-up and the opportunity to clarify the projection step in our method. We will revise our paper with the following detailed modifications:
>
> 1.) **Clarification in the main text:**
> We will expand the explanation about the projection step and the $\texttt{Hardmax}$ operator immediately after introducing projected gradient descent. The updated sentence in lines 182–183 in our paper will read:
>
>
> “We optimize this objective by applying projected gradient descent [27] to estimate the matrix $\boldsymbol{C}$. Specifically, applying ordinary gradient descent steps could deviate from the feasible set $\mathcal{C}$ on the Grassmann manifold. Therefore, we use projected gradient descent to periodically project $\boldsymbol{C}$ back onto the Grassmann manifold after a fixed number of gradient descent steps using the operator $\texttt{Hardmax}(\boldsymbol{C})$. $\texttt{Hardmax}(\boldsymbol{C})$ applies a hard maximum at each row of $\boldsymbol{C}$, ensuring column-wise orthogonality, which coincides with the structural constraints of the Grassmann manifold.”
>
>
> 2.) **Modification of Algorithm 1:**
> We will revise Line 9 in Algorithm 1 to explicitly include the projection step using the $\texttt{Hardmax}$ operator. The revised line in the algorithm will include:
>
> "
> **9.** $\boldsymbol{C}_{t+1} = \texttt{Hardmax}(\boldsymbol{C}_t^*)$ ,
> "
>
> where $\boldsymbol{C}_t^*$ denotes the value of $\boldsymbol{C}$ after a fixed number of gradient steps.
>
> 3.) **Clarification for SINGC in the main text:**
> After introducing the SINGC algorithm and noting that it uses standard gradient descent, we will add the following clarification:
>
> “We note that, for SINGC, we empirically observed that projecting only once at the end of the optimization yields performance similar to using intermediate projections, as done for INGC. Therefore, for simplicity and efficiency, we chose to apply standard gradient descent throughout the iterations and perform the projection using $\texttt{Hardmax}(\boldsymbol{C})$ only at the end.”
>
>
> The above note will be inserted at line 187 in our main text.
>
> We hope this detailed explanation and revision plan address the reviewer’s concern. If there are any further questions, we would be happy to address them.

---

> > ### Comment · Reviewer_9tpS · 2025-08-04
> > **Official comment by Reviewer 9tpS**
> >
> > Thanks for your clarification. I have no concern now. I decide to raise the rating to accept.

---

### Official Review · Reviewer_pbs3 · 2025-06-30

**Clarity:** 3
**Significance:** 2
**Originality:** 2
**Rating:** 4
**Confidence:** 5

**Summary:**

This work presents a graph coarsening method aimed at preserving inner products between node features. A new Inner Product Error (IPE) metric is introduced to measure how the inner products between graph signals are preserved. New connections between IPE minimization and structural consistency of the coarsened graph are established to confirm the effectiveness of IPE. In experiments, both SBM random graph model and real-world graphs are employed to evaluate the proposed INGC model.

**Questions:**

1. Is the proposed method applicable to heterophilic graphs where nearby nodes are likely from different classes?
2. Given a graph Laplacian $L$ and graph signals $X$, Dirichlet Energy considers only the diagonal elements of $X^TLX$, i.e. $tr(X^TLX)$, while the proposed IPE additionally considers off-diagonal elements. **In addition to existing theoretical results in Prop. 3.4**, how to understand the contributions of these off-diagonal elements?
3. How to handle graphs without node attributes?

**Ethical Concerns:**

["NO or VERY MINOR ethics concerns only"]

**Final Justification:**

After discussing with the authors, the major weaknesses identified from my original review still apply, so I decided to maintain the rating. Nevertheless, the overall quality of the work is commendable, and I am inclined to recommend its acceptance.

**Limitations:**

scalablity and applicability to graphs of varying homophily could be discussed.

**Quality:**

3

**Strengths And Weaknesses:**

Strengths:

1. The proposed IPE appears novel.
2. Comprehensive theoretical analyses are provided to explain why IPE minimization is beneficial.
3. The paper is well-written and probably easy to follow.

Weaknesses:

1. The overall contribution is somewhat limited. Apart from the IPE term, the rest of the objective is heavily borrowed from FGC [14]. Notably, the Grassmann manifold-based inner product preservation is a commonly used technique in graph signal processing, so I don't consider it a novel contribution.
2. As described in the first paragraph of the Introduction, the motivation is to speed up graph machine learning by reducing the graph. However, the theoretical time complexity of the proposed method is quadratic in the number of nodes, making it hard to handle large-scale graphs in practice. For small-to-medium scale graphs, there is typically no need to perform graph reduction beforehand. This nature makes the proposed method less meaningful. There are also no experiments conducted on large-scale graphs such as ogbn-products and ogbn-papers100M.
3. The baselines are outdated.

---

> ### Author Rebuttal · Authors · 2025-07-30
>
> Thank you for the time and effort you put into reviewing our paper. Your
> comments were very constructive and helped us improve the manuscript. Our responses to the main concerns you raised are:
>
> # *Contribution and relation to FGC*
>
> Our method proposes a complementary approach to FGC that addresses a different aspect of the coarsening problem. While FGC focuses on preserving individual node features via reconstruction (i.e., minimizing $\|\boldsymbol{X} - \boldsymbol{C} \boldsymbol{X}_c\|$) and promoting smoothness on the coarsened graph (i.e., minimizing $tr(\boldsymbol{X}_c^\top \boldsymbol{L}_c \boldsymbol{X}_c)$), our method aims to preserve the mutual relationships between node features by minimizing the IPE. These objectives are complementary, and in principle, both approaches could be combined into a unified framework.
>
> The diagonal elements of $\boldsymbol{X}^\top \boldsymbol{L} \boldsymbol{X}$ reflect only the individual smoothness of signals/node features (i.e., their Dirichlet energy) with respect to the graph, and ignore the cross-relationships between different signals. In contrast, the off-diagonal elements capture the pairwise interactions between different node features. Specifically, the $(s,k)$-th entry is given by:
> \begin{align*}
> \boldsymbol{X}^\top \boldsymbol{L} \boldsymbol{X}[s,k] = \boldsymbol{x}^{(s)\top} \boldsymbol{L} \boldsymbol{x}^{(k)} = \sum_{(i,j) \in E} w_{ij} \left( x^{(s)}(i) - x^{(s)}(j) \right)\left( x^{(k)}(i) - x^{(k)}(j) \right),
> \end{align*}
> where $\boldsymbol{x}^{(s)} = \boldsymbol{X}[s,:]$ and $\boldsymbol{x}^{(k)} = \boldsymbol{X}[k,:]$ are the $s$-th and $k$-th node feature vectors, respectively. These terms reflect how similarly pairs of features vary across the graph, thus encoding valuable structural information.
>
> Proposition 3.4 establishes a connection between these off-diagonal terms and the Grassmann similarity. Together with Proposition 3.5, they imply that maximizing the Grassmann similarity leads to IPE minimization for smooth signals and improves both the REE and DEE coarsening metrics. This result provides theoretical justification for including the Grassmann term in our objective function.
>
> We note that the only components borrowed from FGC are the regularization terms. These were intentionally reused to ensure a fair and controlled comparison, allowing us to isolate and evaluate the contribution of our proposed IPE term and overall coarsening approach.
>
> Importantly, our formulation is based on Grassmann similarity rather than distance, which enables us to incorporate it directly into a differentiable loss term with a closed-form expression. This choice is key to allowing efficient gradient-based optimization and is central to the novelty of our method.
>
> # *Datasets and baselines*
>
> **Baselines:** We note there exist two recent graph coarsening methods—CMGC [1] and SGBGC [2] that could be considered as baselines; however, their experimental setups differ significantly from ours. Specifically, they adopt a 60/20/20 data split and evaluate classification performance via cross-validation on only 20\% of the labels.
>
> In contrast, we follow the evaluation protocol established in SCAL and FGC, where a Graph Neural Network (GNN) is trained on the coarsened graph and used to predict all node labels for the original graph.
>
> We attempted to include CMGC and SGBGC in our experiments, but were unable to reproduce their code due to missing information (e.g., missing README in SGBGC and broken repository link for CMGC). As a result, we were unable to evaluate their methods in our experimental setup and therefore did not include them in the original comparison.
>
> Following your comment, we conducted an additional search and identified a third recent method, MGC[3], that is compatible with our setting.
> A comparison between our methods and the additional baseline is detailed in the table below.
> We will include MGC as an additional baseline in Table~2 of the revised paper, and our results continue to demonstrate the strong performance of our proposed method.
>
> |   Dataset   |   r  |            MGC[3]           |          INGC (Ours)          |         SINGC (Ours)        |
> |-------------|:----:|:---------------------------:|:-----------------------------:|:---------------------------:|
> | Cora        |  0.3 |  $\underline{84.56\pm1.40}$ |  $\boldsymbol{87.55\pm0.16}$  |        $84.51\pm0.33$       |
> |             |  0.1 |        $76.02\pm0.93$       |  $\boldsymbol{83.38\pm0.47}$  |  $\underline{82.76\pm0.32}$ |
> | Citeseer    |  0.3 |        $74.60\pm1.20$       |  $\boldsymbol{76.89\pm0.23}$  |  $\underline{76.66\pm0.27}$ |
> |             |  0.1 |  $\underline{70.57\pm1.25}$ |         $72.63\pm0.25$        |        $69.71\pm0.72$       |
> | Co-phy      | 0.05 | $\boldsymbol{94.52\pm0.19}$ |   $\underline{94.29\pm0.10}$  |        $94.04\pm0.06$       |
> |             | 0.03 |  $\underline{93.64\pm0.25}$ | $\boldsymbol{94.20 \pm 0.13}$ |       $93.52 \pm 0.13$      |
> | Pubmed      | 0.05 |        $81.89\pm0.00$       |  $\boldsymbol{83.59\pm0.22}$  |  $\underline{83.55\pm0.32}$ |
> |             | 0.03 |        $80.70\pm0.00$       |   $\underline{81.93\pm0.22}$  | $\boldsymbol{83.19\pm0.18}$ |
> | # Best      |      |              1              |               5               |              1              |
> | # 2- Best   |      |              3              |               2               |              3              |
> **Table 2.** Node classification accuracy across datasets and coarsening ratios ($r$). Best results are in bold; second-best results are underlined. The last two rows indicate the number of times each method achieved the best and second-best performance.
>
> **Hetrophily graphs:**  A central assumption in our work is that node features are smooth with respect to the graph structure, which is connected to the graph homophily assumption. This smoothness assumption underlies both our formulation and theoretical guarantees. For this reason, we do not include heterophilic datasets in our experimental evaluation.
>
> More specifically, the second term in our objective relies on this smoothness assumption, as it encourages alignment with the subspace spanned by the low-frequency eigenvectors of the graph Laplacian. In heterophilic graphs, this assumption does not hold, and our theoretical results would no longer be valid.
>
> One potential direction for adapting our method to such settings is to modify the second term in the objective to project onto a different frequency range—e.g., using $\boldsymbol{U}^{(k)}$ corresponding to mid- or high-frequency eigenvectors if prior knowledge of label-frequency alignment is available. However, such a modification would require a fundamentally different theoretical treatment, which we believe is beyond the scope of this work.
>
> Following your suggestion, we will add a discussion of this limitation and possible future extensions to the Conclusion section of the revised manuscript.
>
> # *Scalability*
> We acknowledge that optimization-based coarsening methods—including ours and FGC—have scalability limitations when applied to extremely large graphs. A commonly adopted strategy to address this issue is to partition the graph into smaller, manageable subgraphs[4], process each independently (and potentially in parallel), and then reconnect them.
>
> We will clarify this limitation in the updated paper and explain how such a strategy can be incorporated into our framework to extend its applicability to large-scale graphs.
>
> # *Additional Clarification*
>
> Please note that our second proposed algorithm, SINGC, does not use node attributes as part of its objective; its optimization is performed solely based on the graph structure via the eigenvectors of the Laplacian $\boldsymbol{U}^{(k)}$. As a result, it is directly applicable to graphs without node features.
>
>
>
> [1] Dickens et al. "Graph coarsening via convolution matching for scalable graph neural network training." Companion Proceedings of the ACM Web Conference 2024.
>
> [2] Shuyin Xia et al. "Graph Coarsening via Supervised Granular-Ball for Scalable Graph Neural Network Training" The Thirty-Ninth AAAI Conference on Artificial Intelligence (AAAI-25).
>
> [3] Halder et al. "Multi-Component Coarsened Graph Learning for Scaling Graph Machine Learning" Companion Proceedings of the ACM on Web Conference 2025.
>
> [4] Karypis et al. "A fast and high quality multilevel scheme for partitioning irregular graphs." SIAM Journal on scientific Computing 20.1 (1998): 359-392.

---

> > ### Comment · Reviewer_pbs3 · 2025-08-03
> >
> > Dear authors, thank you for the response. I have no further questions and have decided to maintain my original rating.

---

### Official Review · Reviewer_XzK6 · 2025-07-02

**Clarity:** 3
**Significance:** 3
**Originality:** 3
**Rating:** 4
**Confidence:** 3

**Summary:**

This paper proposes the inner product error as a loss function for evaluating the quality of a graph coarsening. In particular, for a graph Laplacian $L$ with node features $X$ and a coarsened Laplacian $L_c$ with coarsened features $X_c$, the proposed inner product error is defined $\| X^TLX - X_c^TL_cX \|_F^2$. The authors further propose the INGC algorithm for graph coarsening. INGC finds a coarsening by solving a constrained optimization problem (Equations 8 and 9). The loss function includes the inner product error between the original graph and the coarsened graph, as well as smoothness conditions on the coarsening and regularization terms aimed at making the coarsening well-defined and the coarsened graph connected. The paper also presents some theoretical justfications of the inner-product loss, as well as experiments on several benchmark datasets.

**Questions:**

## Questions

(1) The statement of Proposition 3.2 requires a certain condition (the graph has n-k connected components), but I cannot see where this condition is used in the proof. Why is this condition needed?

(2) Does the runtime of INGC in Table 3 in Supplement account for the time to do the projection?

## Suggestions

- I believe for the properties of Equation (7) to hold, it needs to be the case that P_{i,j} > 0 if *and only if* j is mapped to the ith super node above. The way it is written in your paper, only one of these conditions is required for P. (I am basing this off the proof of Proposition 6 in [Loukas, 2020])

- The proof environments seem a little weird. I would suggest to use the \begin{proof}\end{proof} commands that are a part of amsthm.

## Typos

- Line 148: rank(L) < k

- Proposition 3.2: The phrasing "for any two graph signals" is ambiguous, as "for any" could either mean for all or there exists. I would recommend changing this to "for all pairs of graph signals".

- Proposition 3.2: In this main text, this result is called a "Proposition", but in the appendix, it is called a "Theorem". Likewise, Line 209 refers to Proposition 3.4 as "Theorem 3.4".

- Appendix, Line 818: The LaTeX commands \log and \det should be used.

**Ethical Concerns:**

["NO or VERY MINOR ethics concerns only"]

**Final Justification:**

I appreciate the authors' responses. Nevertheless, I think the main weakness I listed in the original reviews stand and I will maintain my rating.

**Limitations:**

Yes

**Quality:**

3

**Strengths And Weaknesses:**

## Strengths

- The inner-product loss makes a lot of sense. The authors use theoretical results to justify the two main components of their loss terms. Propostions 3.5 show that minimizing the proposed loss also minimizes the losses REE and RE (definitions 2.1 and 2.2), two existing losses from the graph coarsening literature. While Proposition 3.2 and 3.4 don't hold in all cases, Proposition 3.5 provides an analysis that works in all cases and that gives better theoretical results the lower the loss. Moreover, the authors perform a simple experiment (table 5) to show the conditions of Proposition 3.5 are realistic.

- The proposed coarsenings outperform existing coarsening techniques on several datasets and on several evaluation metrics (Table 1).


## Weaknesses

- While the proposed inner-product loss is new and well-motivated, its contribution appears a llittle incremental. For the theoretical results, the conditions for Proposition 3.2 are too strong, and Proposition 3.4 is not surprising.  (Proposition 3.5 is the most interesting result of the paper in my view.)

- While Table 1 shows better graph coarsening metrics of proposed methods, the advantage in node classification is not large (Table 2). At the same, the time to compute INGC and SINGC can be quite a bit slower than existing method FGC – see Table 4 in the Supplement – even though they have similar asymptotic runtimes.

---

> ### Author Rebuttal · Authors · 2025-07-30
>
> Thank you for the time and effort you put into reviewing our paper. Your
> comments were very constructive and helped us improve the manuscript. Our responses to the main concerns you raised are:
>
> # *Theoretical results clarifications*
>
> We thank the reviewer for their valuable comment on Proposition~3.2. We acknowledge that the rank condition stated in the proposition might be too strong and is not required for the proof. It is sufficient to state that if the inner product is preserved for all pairs of graph signals, i.e., $\boldsymbol{x}^\top \boldsymbol{L} \boldsymbol{y} = \boldsymbol{x}_c^\top \boldsymbol{L}_c \boldsymbol{y}_c$, then the original Laplacian $\boldsymbol{L}$ can be exactly recovered from $\boldsymbol{L}_c$. However, a necessary condition for the first identity to hold is that the rank of $\boldsymbol{L}$ is at most $k$; otherwise, it is not feasible to recover $\boldsymbol{L} \in \mathbb{R}^{n \times n}$ from $\boldsymbol{L}_c \in \mathbb{R}^{k \times k}$.
>
> Following your suggestion, we will revise the paper and remove the rank assumption from the proposition statement and instead include it as a remark immediately following the result. We will also clarify that, although this condition is unrealistic in most practical settings, the result is not intended to imply that exact reconstruction is feasible in general. Rather, it is meant to theoretically motivate our focus on minimizing inner product error as a meaningful proxy for preserving the graph’s structure during coarsening.
>
> Although the results of Proposition~3.4 may not be surprising, the proposition supports the development of our method. It formally establishes the connection between minimizing the IPE and maximizing the Grassmann similarity between the coarsening operator $\boldsymbol{C}$ and the top-$k$ eigenvectors of the original Laplacian, $\boldsymbol{U}^{(k)}$.
> In particular, it shows that maximizing Grassmann similarity leads to IPE minimization for all graph signals that satisfy the smoothness assumption, enabling a principled method that generalizes to unseen smooth signals as well.
>
> This connection is also crucial for understanding the implications of Proposition~3.5, as it provides a theoretical bridge: minimizing IPE for general smooth signals implies improvements in both REE and DEE metrics.
>
> Importantly, our formulation is based on Grassmann similarity rather than distance, which enables us to incorporate it directly into a differentiable loss term.
>
> # *Method and run time clarification*
>
> The runtime results in Table~3 include all major computational steps: spectral decomposition, optimization, and the final projection of the features and Laplacian, i.e., computing $\boldsymbol{X}_c = \boldsymbol{P} \boldsymbol{X}$ and $\boldsymbol{L}_c = \boldsymbol{C}^\top \boldsymbol{L} \boldsymbol{C}$.
>
> We acknowledge that INGC is slower compared to other methods, which is expected since it considers richer structural information by preserving pairwise feature relationships. In contrast, SINGC is significantly more efficient, as its main computational cost arises from the eigenvalue decomposition, while the optimization step is relatively efficient. We believe this trade-off is justified given the consistent improvement in coarsening quality.
>
> It is important to note that our approach is not adversarial to FGC. While FGC focuses on reconstructing individual node features and enforcing smoothness on the coarsened graph, our method aims to preserve the mutual relationships between node features by minimizing the IPE. These objectives are complementary, and in principle, both approaches could be integrated into a unified framework. However, to clearly isolate and demonstrate the advantages of our proposed perspective, we evaluate it independently of FGC in this work, using the same regularization terms.
>
> # *Other suggestions and Typos*
> Thank you for pointing out our formatting issues. We accept all your suggestions and will fix them in the revised version of the paper.

---

> > ### Comment · Reviewer_XzK6 · 2025-08-03
> >
> > Thanks for the response. I don't have further questions

---

### Decision · Program_Chairs · 2025-09-17

**Decision:**

Accept (poster)

**Comment:**

The paper introduces a new loss for graph coarsening, the Inner Product Error (IPE), that aims to preserve node features during the coarsening process. The paper also develops two algorithms, INGC and SINGC, which minimize IPE while incorporating Grassmann alignment with Laplacian eigenvectors, showing consistent improvements in coarsening metrics and competitive node classification performance.

Reviewers particularly appreciated the theoretical motivation for the use of the IPE loss provided by the authors. At the same time, after the rebuttal, some concerns about the assumptions behind the theoretical analysis remain unresolved. In particular, the reviewers are concerned about the reliance on feature smoothness, which restricts applicability to homophilic graphs and leaves heterophilic settings unaddressed. Reviewers also noted that the spectral decomposition might be a bottleneck on larger graphs.

Overall, while not without limitations, the paper presents a technically sound and reasonably novel perspective on graph coarsening, with clear theoretical motivation and empirical support.